



# Hydrostratigraphic modelling using multiple-point statistics and airborne transient electromagnetic methods

Adrian A.S. Barfod[1,2], Julien Straubhaar[4], Anne-Sophie Høyer[1], Júlio Hoffimann[3], Anders V. Christiansen[2], Ingelise Møller[1], Jef Caers[3]

[1]Department of Groundwater and Quaternary Geology Mapping, Geological Survey of Denmark and Greenland (GEUS), C.F. Møllers Allé 8, 8000 Aarhus C
[2]Hydrogeophysics group, Department of Geoscience, Aarhus University, C.F. Møllers Allé 4, 8000 Aarhus C
[3]Stanford Center for Reservoir Forecasting, School of Earth, Energy & Environmental Sciences, Stanford University, Green Earth Sciences, 367 Panama St, Stanford, CA 94305
[4]Centre d'Hydrogéologie et de Géothermie (CHYN), Université de Neuchâtel, Switzerland

Correspondence to: Adrian A.S. Barfod (adrian.s.barfod@gmail.com)

**Abstract.** Creating increasingly realistic hydrological models involves the inclusion of additional geological and geophysical data in the hydrostratigraphic modelling procedure. Using Multiple Point Statistics (MPS) for stochastic hydrostratigraphic modelling provides a degree of flexibility that allows the incorporation of elaborate datasets and provides a framework for stochastic hydrostratigraphic modelling. This paper focuses on comparing three MPS methods: *snesim*, *DS* and *iqsim*. The MPS methods are tested and compared on a real-world hydrogeophysical survey from Kasted in Denmark, which covers an area of 45 km$^2$. The comparison of the stochastic hydrostratigraphic MPS models is carried out in an elaborate scheme of visual inspection, mathematical similarity and consistency with boreholes. Using the Kasted survey data, a practical example for modelling new survey areas is presented. A cognitive hydrostratigraphic model of one area is used as Training Image to create a suite of stochastic hydrostratigraphic models in a new survey area. The advantage of stochastic modelling is that detailed multiple point information from one area can be easily transferred to another area considering uncertainty.

The presented MPS methods each have their own set of advantages and disadvantages. The *DS* method had average computation times of 6-7 h, which is large, compared to *iqsim* with average computation times of 10-12 min. However, *iqsim* generally did not properly constrain the near-surface part of the spatially dense soft data variable. The computation time of 2-3 h for *snesim* was in between *DS* and *iqsim*. The *snesim* implementation used here is part of the Stanford Geostatistical Modeling Software, or SGeMS. The *snesim* setup was not trivial, with numerous parameter settings, usage of multiple grids and a search tree database. However, once the parameters had been set it yielded comparable results to the other methods. Both, *iqsim* and *DS*, are easy to script and run in parallel on a server, which is not the case for the *snesim* implementation in SGeMS.





## 1 Introduction

Recent advances in hydrology have shown the importance of accurate hydrologic models for management of increasingly sparse groundwater resources. Hydrologic modelling predictions are sensitive to geologic heterogeneity (e.g. Freeze 1975, Gelhar 1984, Fogg et al. 1998, LaBolle and Fogg 2001, Zheng and Gorelick 2003, Feyen and Caers 2006, Fleckenstein et al.
2006, Zhao and Illman 2017). However, geological units include complexities not directly related to hydrofacies (Klingbeil et al. 1999). Instead the concept of *hydrostratigraphic units* is used throughout this study, which effectively combines geological units and reduces the total number of units resulting in a closer relation to the hydrologic units. Improving the realism and quantification of uncertainty around hydrostratigraphic models is therefore an important step towards accurate groundwater modelling predictions. Hydrostratigraphic models are created using several approaches. A common approach is
a manual co-interpretation of available geophysical, geological and/or hydrologic data. The geoscientist cognitively uses his/her refined knowledge of geological processes combined with the provided datasets to create a detailed cognitive geological model (e.g. Jørgensen et al. 2013). The cognitive geological model is then simplified to a hydrostratigraphic model. Even though the hydrostratigraphic model encapsulates the complexities related to geologic architecture, it does not reflect the hydrostratigraphic uncertainty. It is a so-called deterministic model, *i.e.* one version of the hydrostratigraphic
subsurface. An alternative to cognitive modelling is stochastic modelling using geostatistical methods. The field of geostatistical modelling focuses on creating models depicting subsurface hydrogeology and/or reservoir properties. Geostatistics is currently applied in a number of geoscience fields, such as petrology (e.g. Okabe and Blunt 2005), petroleum reservoir modelling (*e.g.* Journel and Zhang 2006, Strebele et al. 2002), hydrogeology (e.g. Huysmans and Dassargues 2009), hydrology (*e.g.* Michaelides and Chappell 2009). Overall geostatistical methods provide a framework in which
multiple equiprobable hydrostratigraphic models can be created in a semi-automated fashion. The individual stochastic models do not reflect the modelling uncertainty, but the model ensemble does. The multiple hydrostratigraphic models can be used as a set of input parameters for the groundwater model. By running the groundwater model several times with different hydrostratigraphic models, multiple predictions can be made, yielding an estimate of the prediction uncertainty. The ability to understand how the hydrostratigraphic uncertainty is related to the prediction uncertainty will help in understanding
where to improve the hydrostratigraphic models in order to reduce the prediction uncertainty. This study will however not focus on groundwater modelling predictions, but on the presentation of a stochastic modelling framework for reconstructing subsurface hydrostratigraphic architecture.

Today *state-of-the-art* geostatistical tools are readily available to geoscientists. Traditional two-point statistics, or variogram based methods, *e.g. sisim (Journel 1983)* and *sgsim* (Deutsch and Journel 1998), have been widely used in both research and
in practice (*e.g.* Seifert and Jensen 1999, Caers 2000, Juang et al. 2004, Delbari et al. 2009). However, variogram based techniques depend on two-point statistics for simulation of complex geological features. Depending on the complexity of the geological setting, such two-point statistical methods cannot re-create complex curvilinear geological features of the subsurface which are common in fluvial and glaciofluvial environments (e.g. Arpat and Caers 2005, Hu and Chugunova



2008, Journel and Zhang 2006, Journel 1993, Liu 2006, Sánchez-Vila et al. 1996, Strebelle and Journel 2001). An additional

geostatistical modelling tool which should be mentioned is T-PROGS (Carle 1999). T-PROGS is based on transition probabilities between categories and generates geostatistical realizations based on such constraints. In comparison with indicator method, *sisim*, it allows for better integration of these transition probabilities and hence, the spatial cross-correlations of soil/rock type architecture into the groundwater models. However, T-PROGS also has difficulties in re-constructing curvilinear geological features. Kessler *et al.* (2013) made a detailed comparison between T-PROGS

realizations and real-world cross-sections in a gravel pit in Denmark. The result reveals a suboptimal pattern reproduction, in comparison to other simulation tools such as Multiple-Point statistics (MPS) (Mariethoz and Caers 2014b). MPS is a recent alternative to classic two-point statistics. Here, additional multiple-point (MP) information from a training image (TI) is used to condition the simulations. The usage of MP information allows for reconstruction of more complex geological features, such as curvilinear features (Strebelle 2002). A TI is any 2D/3D image containing geometrical information relevant to the

hydrostratigraphic model. The crux of the MPS approach is finding a relevant TI. Some examples of 2D/3D TIs are: categorical images of outcrops (2D), categorical drawings of a geological system created by a geoscientist (2D), cognitive geological or hydrostratigraphic voxel models (3D) (e.g. Høyer et al. 2015a) *etc*. Today, MPS techniques are widely used in geoscientific research and studies, a few examples are: Maharaja (2005), Meerschman *et al.* (2013), Hermans *et al.* (2014). The MPS framework allows for conditioning of geological architecture/patterns, a stochastic framework and spatially

constraining to both soft data and hard data (Arpat and Caers 2005, Guardiano and Srivastava 1993, Journel 1993, Strebelle and Journel 2001).

Within the geostatistics framework the creation of hydrostratigraphic models requires the inclusion of data from multiple sources, often geophysical models (soft data), borehole data (hard data) and a TI. The different data sources each provide relevant information. Geophysical models provide information regarding the large-scale hydrostratigraphic architecture.

Boreholes, on the other hand, provide detailed yet usually sparse information regarding hydrostratigraphic units. Each data source is a piece of the puzzle, combining the individual pieces improves the resulting hydrostratigraphic models. The inclusion of several types of data is, however, not trivial since information regarding their mutual relationships, *e.g.* the hydrostratigraphic-petrophysical relationship, is required. An important source of information which helps to combine the different sources of data is geologic knowledge. Geologic knowledge can be defined as information regarding geologic

processes, geomorphologic patterns, structural geology *etc*. Incorporating geological knowledge into hydrostratigraphic models is often difficult and done *ad-hoc*. Geologic information, as described above, complements the soft data and helps to create more realistic hydrostratigraphic models. However, within the MPS framework this type of information can be implemented *via* the TI.

This study focuses on comparing and testing three MPS methods on a real-world dataset from a groundwater survey in

Kasted, Denmark. An important part of the dataset is the airborne geophysical survey, providing a set of resistivity models containing information regarding the large-scale hydrostratigraphic architecture of the area. The MPS tools are used to



reconstruct an intricate system of interconnected buried valleys. The end result is an ensemble of hydrostratigraphic models. A 3D hydrostratigraphic voxel model of the area is used as a TI, containing detailed MP information regarding the hydrostratigraphic features of the survey area. Information regarding the geological architecture and the relationship between

hydrostratigraphy and petrophysical properties are contained in the TI. The hydrostratigraphic-petrophysical relationship is explicitly known since the hydrostratigraphic model spatially overlaps with the geophysical and borehole lithology logs. Spatially constraining the simulation to the soft data, consisting of the resistivity models, ensures that simulated geological patterns are placed concurrently to the real-world. However, such geophysical soft data have several types of related uncertainty, *e.g.* spatial uncertainty related to incomplete datasets, resolution capabilities, signal-to-noise ratio decrease with

depth *etc*. Incomplete geophysical datasets is a common problem and are typically reconstructed using geostatistics; often in a deterministic fashion. A common approach is to use variogram based geostatistics, such as Kriging interpolation, to reconstruct the incomplete resistivity grid (Isaaks and Srivastava 1989). We have used the stochastic direct sampling (*DS*) grid reconstruction routine proposed by Mariethoz and Renard (2010). Here, the grid reconstruction uncertainty is reflected by multiple resistivity grids, yielding variable patterns in the multiple reconstructed grids. The reconstructed grids are then

used in conjunction with the hydrostratigraphic TI to create a set of stochastic hydrostratigraphic realizations of the hydrostratigraphy of the modeled area.

In relation to the Danish Groundwater mapping campaign (Thomsen et al. 2004) detailed geophysical datasets (Møller et al. 2009) and hydrostratigraphic models exist. A selection of the 3D geologic and hydrostratigraphic voxel models are reported in the literature, *e.g.* Høyer *et al*. (2015a), Høyer *et al*. (2015b) and Jørgensen *et al.* (2015). Additionally, the study by Høyer

*et al.* (2016) presents a framework for making large-scale MPS models based on geological 3D voxel models, and seismic and borehole data. In this study, we will show a practical application where an existing cognitive model from an area is used as a TI for simulating a new survey area with similar geological characteristics.

To our knowledge, no vigorous studies comparing multiple MPS methods have been carried out on real-world hydrogeophysical datasets. By applying several measures to assess and compare the modeling results, the selected MPS tools

are tried, tested and compared on real-world data. The main contributions of this study are: 1) a practical real-world example of stochastic reconstruction of incomplete geophysical datasets, 2) Comparison of three MPS methods for integrating geophysical data: *snesim* (Liu 2006, Strébelle and Journel 2000), direct sampling (*DS*) (Mariethoz et al. 2010) and image quilting (*iqsim*) (Hoffimann et al. 2017, Mahmud et al. 2014), 3) validation of the comparison results by: a) visual inspection, b) a mathematical comparison method called the "analysis of distance" (ANODI) (Tan et al. 2014), c)

comparison of the simulation results against the borehole lithology logs, 4) to show the strengths/weaknesses of a stochastic hydrostratigraphic modelling framework, and 5) a practical example showing how to use the cognitive hydrostratigraphic interpretation of one area to directly generate hydrostratigraphic models of new areas, using only the soft data from the new area.





## 2 Study area and data

The Kasted survey area is located in Denmark, in the eastern part of Jutland, close to the city of Aarhus (Figure 1a). The 45 km$^2$ area has been surveyed in detail and contains 453 boreholes; a SkyTEM survey of 333 line km. A detailed geologic model of the area has been created by Høyer *et al.* (2015a). The dataset was collected and compiled in relation to the HyGEM project. The local geology consists of an intricate system of interconnected Quaternary buried valleys, infilled with till and meltwater deposits. The buried valleys are incised into thick hemipelagic Paleogene clay which dominates the area

(Høyer et al. 2015a). Many such pre-glaciated areas are dominated by buried valleys, which have proven important subsurface features in regard to groundwater flow (Jørgensen and Sandersen 2006, Seifert et al. 2008). These noteworthy geological features have received a lot of attention in research through the years (e.g. Destombes et al. 1975, Jørgensen and Sandersen 2009, Kehew et al. 2012, Ritzi et al. 1994)

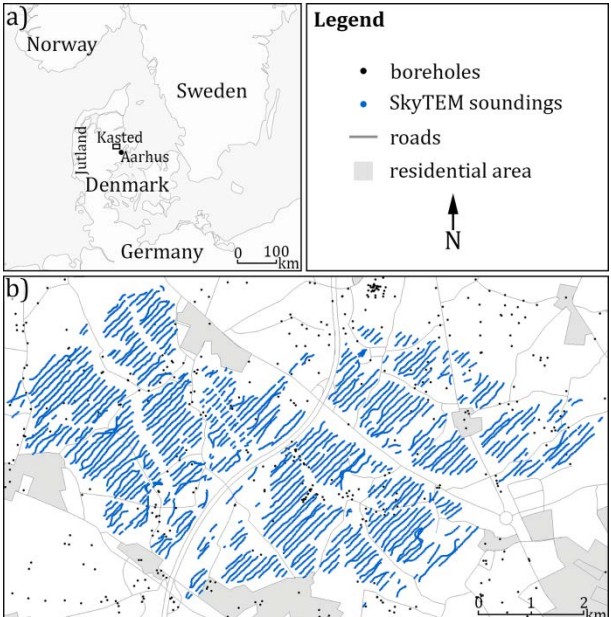

**Figure 1: An overview map of the Kasted survey area; a) shows the geographic location of the survey area which is marked as a black box, and b) shows a close-up view of the Kasted survey area with the related datasets and infrastructure overlay.**

The dataset used in this study consists of a dense airborne geophysical SkyTEM survey, near-surface boreholes from the Danish borehole database, and a cognitive geologic model created by an experienced geoscientist. In the following we will summarize the key features of these datasets.

The SkyTEM system (Sørensen and Auken 2004), is a helicopter Transient Electromagnetic system allowing for rapid collection of large geophysical datasets, with high spatial density. The Kasted SkyTEM survey contains 333 line km with a line spacing of roughly 100 m (Figure 1b). The SkyTEM data are inverted and modeled according to the scheme described by Viezzoli *et al.* (2008), the end result being a collection of spatially constrained inversion models. In Denmark it is





standard protocol to calibrate the SkyTEM system at an official calibration site, as described by Foged et al. (2013), ensuring

data of high-quality and reproducible results. Therefore, the resistivity values from a calibrated SkyTEM survey are comparable to other calibrated SkyTEM surveys. The SkyTEM system is sensitive towards large-scale conductive trends in the subsurface, especially when a significant contrast between a conductive and a resistive feature exists. In the eastern part of Jutland it is common that the lower confining boundaries of the buried valleys are well resolved since these buried valleys are often quite resistive and are eroded into conductive hemipelagic Paleogene clays.

The Danish borehole database, JUPITER (Hansen and Pjetursson 2011), contains about 280.000 shallow boreholes which have been drilled for a variety of purposes, mainly in relation to drinking water, and raw materials exploration, but also in relation to research, and geotechnical studies. The JUPITER database contains information on location, drilling method, lithology, geologic age, filter position, water chemistry, *etc*.

The cognitive geologic model was created using all available data, including the 333 line km of SkyTEM data, information

from 435 boreholes and prior geological knowledge of the area. The model was created using the cognitive modelling scheme which is introduced by Jørgensen et al. (2013). The geological model is described in great detail by Høyer et al. (2015a). The geologic model is detailed and contains a set of 21 interconnected buried valleys. The final 3D voxel model contains 42 unique geological units, which are simplified into three overall hydrostratigraphic units in this study. The three hydrostratigraphic units are chosen for the purpose of covering the overall hydrological features of the groundwater

modelling area. The cognitive hydrostratigraphic model will act as a base-line for assessing the performance of the three MPS methods, and the stochastic modeling results will be compared against the cognitive model.

## 3 Methods

MPS provides a degree of flexibility which assists the modeler in creating geologically realistic hydrostratigraphic models. The idea is to create a suite of hydrostratigraphic models which span a realistic subset of possible model architectures, as

opposed to a deterministic model which spans a single possible model architecture. The term realistic refers to models which comply with the underlying data sets mentioned above, *i.e.* borehole lithological logs, geophysical resistivity models, and the cognitive geological model. The underlying datasets have associated uncertainties describing ranges of possible models. The suite of equiprobable hydrostratigraphic models can be used as input to a groundwater model, making it straightforward to test the sensitivity of specific groundwater model predictions.

### 3.1 MPS methodologies

MPS methods use a training image (TI) to condition a model simulation to a prior geological conceptualization. As opposed to two-point statistics, the joint variabilities of multiple-points (MP) are assessed at the same time during simulation. The MP joint variabilities cannot be inferred from sparse data and are therefore taken from a relevant exhaustive TI. The justification that a given TI can be used to infer the joint variability of MPs heavily lies on the choice of a relevant TI. A TI should



always contain geologically realistic and relevant information (Journel and Zhang 2006). Finding and/or creating a realistic TI is thus important to the MPS methodology. A TI is essentially any categorical or continuous image which contains the geological conceptualization of the target variable (Mariethoz and Caers 2014a). It is not a subsurface model itself, but a quantitative conceptual depiction of it. The user chooses the TI based on his/her prior understanding of the local hydrogeological system. The TI does not need to carry any locally accurate information, *i.e.* it does not need to contain the

actual geographical positions of the hydrostratigraphic architecture, just the general patterns. It needs to reflect a prior geological- or structural concept (Strebelle and Journel 2001).

The MPS methods chosen in this study have been selected to reflect recent advances in MPS methods. The MPS methods in this study include: the "single normal equation simulation" (*snesim*) (Strébelle and Journel 2000) implemented in the Stanford Geostatistical Modeling Software (SGeMS), "direct sampling simulation" (*DS*) (Mariethoz et al. 2010)

implemented in *DeeSse* (Straubhaar 2011), and "image quilting simulation" (*iqsim*) (Hoffimann et al. 2017) implemented in ImageQuilting.jl.

### 3.1.1 Single normal equation simulation - *snesim*

The *snesim* method is a traditional MPS method. It fits into the so-called "probability framework" where geophysical models (*not data*) are considered soft information, and as such needs to be converted into probabilities. Suppose we have a

categorical random variable S which has $K$ possible states ($s_k$, $k = 1, ..., K$), *i.e.* there are $K$ hydrostratigraphic units. For each cell in the target sampling grid a probability $prob\{s_k\}$ is defined for each of the $K$ states, so that for a given grid cell, denoted $cell_i$:

$$prob\{cell_i\} = \sum_{k=1}^{K} prob\{s_k\} = 1, \tag{1}$$

where $i \in \{1, ..., N\}$, and the sampling grid has a total of N cells. The crux is then to translate the geophysical data into the

probabilities described in equation (1). The collection of all probabilities for the entire sampling grid is also referred to as a probability map (2D) or probability grid (3D). The translation of the soft data is usually carried out based on a prior understanding of the petrophysical-hydrostratigraphic relationship, and will be discussed further later in the paper. For a detailed description of the more general petrophysical-lithological relationship the reader is referred to *e.g.* Barfod *et al.* (2016) and Beamish (2013). The probability grids are used to constrain the simulation using the so-called tau model (Journel

2002). The probability grid approach is intuitive, and allows the modeler to incorporate any desired datasets or variables into the probability map. Examples of soft data are any type of geophysical soft data and/or prior information which can be translated into probabilities.

In *snesim* the TI is stored in a dynamic data structure called a search tree. The search tree is a database and can be seen as a condensed summary of the full TI. It contains the spatial information to which the simulation is conditioned; for more detail

see Strebelle (2002). To avoid repetitive scanning of the TI, which is computationally expensive, the TI is stored in a search





tree database ahead of the simulation (Roberts 1998). This is done once. TI patterns can then be retrieved from the database without scanning the entire TI. Depending on the amount of detail stored in the search-tree this can be quite CPU intensive, since the entire search tree is stored in memory, and therefore there is an upper limit to the size of the search-tree pattern database. However, advances in computers have increased the upper limit for available CPU.

Another caveat of *snesim* is the usage of multiple-grids (Tran 1994). Due to limitations in relation to the search neighborhood, the simulation of structures on all scales requires the usage of multiple grids. The simulation is carried out on a series of multiple simulation grids with varying density, ensuring pattern reproduction at all scales. The search tree formulation and multiple grid approach add to the overall complexity of parameterization in *snesim*, but at the same time ensure stable and reliable MPS modeling results. The increased number of user defined parameters makes it less intuitive,

since it is relatively difficult to determine the optimal parameter values for a given dataset.

### 3.1.2 Direct sampling simulation - *DS*

The Direct Sampling Simulation (*DS*) method consists, for the simulation of each cell, in randomly scanning the TI until a pattern similar to the pattern centered at the simulated cell is found, and then in copying the value in the center of the pattern from the TI to the simulation grid. As a consequence, contrary to *snesim*, no probability is explicitly computed to draw a

value at a simulation grid cell. In this paper we use the *DeeSse* implementation of *DS,* presented by Straubhaar (2011). This bypasses the necessity of saving spatial patterns in a search tree database; instead spatial patterns are conditioned by directly scanning the TI.

One issue which needs to be solved is how to constrain a soft data variable. In *DS* this is accomplished by introducing an auxiliary variable. The auxiliary variable is roughly a translation of the TI into a soft data variable. Suppose a forward

operator, denoted by $G$, represents the physical model which translates the subsurface hydrostratigraphic units into the continuous soft data variable, as when scanning the near surface with a geophysical instrument and subsequently process and interpret the data into the actual petrophysical parameter. Then we can define an approximate forward operator $G^*$ (Mariethoz and Caers 2014b). The $G^*$ operator is an operator which is used to translate the TI into a spatially overlapping soft data variable. However, in practice creating a $G^*$ operator requires several steps. Based on the modeling setup of this

study, we will briefly review the required steps. Firstly, the TI needs to be populated with relevant resistivity values. The resulting populated resistivity grid does, however, not reflect the physical model, G, which translates the subsurface hydrostratigraphic units into subsurface bulk resistivity. To properly reflect the G operator additional complexity needs to added, such as: smooth layer boundaries, loss of resolution with depth, limited resolution capabilities, the instrument footprint *etc*. This can be achieved by using either an approximate 1D or a full 3D forward modeling code to translate the

populated resistivity models into synthetic data reflecting actually measured field data. These data, the forward responses then need to be processed and inverted back to resistivity models, which now constitute an auxiliary variable which reflects the complexities involved with the SkyTEM system. The auxiliary variable and the categorical hydrostratigraphic variable





are combined to create a multivariate, or bivariate TI. The bivariate TI consists of a categorical variable, *e.g.* the three hydrostratigraphic units, and the geographically overlapping continuous auxiliary variable, representing the soft data

variable. The setup used in this paper, avoids the usage of the $G^*$ operator to create the auxiliary variable, since the reconstructed resistivity grids and cognitive hydrostratigraphic model grids geographically overlap. The reconstructed resistivity grids can thus directly be used as an auxiliary variable for the cognitive hydrostratigraphic model TI. The bivariate TI constituted of collocated categorical hydrostratigraphic units (cognitive model / primary variable) and resistivity values (auxiliary variable) contains information regarding the relationship between these variables. The simulation is then

conditioned against the bivariate TI by using a so-called distance measure. Distance measures are designed to compare the similarity of two sets of spatial patterns to each other. The idea is that similar patterns have relatively small distances, while dissimilar patterns have relatively large distance values. Conditioning against the *MP* information contained in the bivariate TI enables the ability to find probable spatial patterns, which also agree with the soft data variable.

*DS* is more flexible than traditional MPS methods, such as *snesim*. As no search-tree database is required, the multiple grid

formulation used in *snesim* is not required in *DS*, which effectively reduces the number of parameters and makes the parametrization relatively simple. Furthermore, one can simulate continuous variables, and/or discrete variables with no limitation to the maximum number of categories (*e.g.* hydrostratigraphic units). In our case, any number of geophysical datasets collocated or not, can be included as long as a corresponding auxiliary variable is added to the multivariate TI. However, it can be a cumbersome process generating the auxiliary variable. Furthermore, it is even possible to use

probability grids in place of the actual soft data variable, as in *snesim*, if desired (Mariethoz et al. 2015). Depending on the setup and dataset, *DS* can be computationally as fast as *snesim*. Moreover, the *DS* implementation used in this work is amenable to scripting yielding the possibility of improving computation times on computer clusters or servers, if available.

### 3.1.3 Image quilting simulation - *iqsim*

The image quilting simulation (*iqsim*) method has been borrowed from the computer vision literature (Efros and Freeman

2001). The algorithm is originally designed to synthesize and/or replicate patterns from 2D images, but has since been modified to accommodate conditioning data and 3D geoscience problems (Mahmud et al. 2014). The concept of the *iqsim* method is straightforward. In essence, *iqsim* cuts the TI into user defined patches or blocks, and then reassembles the patches to create a simulation. The difficult part is how to re-assemble the patches, to create meaningful and seamless realization results which can be constrained to a soft data variable. These difficulties have been solved, and for more detail see *e.g.*

Hoffimann *et al.* (2017)[1]. A great advantage of the *iqsim* method is its computation time. It has a similar setup to *DS*, regarding the usage of auxiliary variables. The *iqsim* method is new within the field of groundwater and environmental

---

[1] Software is available at https://github.com/juliohm/ImageQuilting.jl.





modelling, and for this paper the open-source Julia implementation by Hoffimann *et al.* (2017) is utilized. So far, this code contains the ability to use masked grids, *i.e.* grids where only specified grid cells are simulated, conditioning hard and soft data, and running simulations on the computer *Graphics Processing Unit* (GPU), yielding computationally fast simulation of hydrostratigraphic models on a personal computer. As with *DS*, there are no limitations to the number of data events, since the search-tree structure is avoided, no multiple-grids are required, effectively making for a simple parameterization.

### 3.2 Reconstructing incomplete dense geophysical datasets

A common problem in hydrogeophysics is that datasets, albeit spatially dense, do not cover the entire modelling grid. In electromagnetic methods human infrastructure causes electromagnetic interference with the signal. Such noisy soundings, referred to as coupled soundings, are removed during processing, as presented by Auken *et al.* (2009), resulting in an incomplete dataset with gaps scattered throughout the survey area (Figure 1b). Several approaches to manage with incomplete datasets exist. One approach is to leave the incomplete dataset as is; meaning gaps are reconstructed during simulation of the hydrostratigraphic model without spatially constraining the simulation gaps. The gaps are filled out solely by conditioning to the TI. Alternatively, dataset gaps can be filled prior to simulation, which is primarily done if the dataset has a high spatial density and/or the underlying random variables describing the data are not assumed to be especially complicated. The soft data utilized for constraining in this study are SkyTEM models. The raw SkyTEM data undergo processing and inversion (Auken et al. 2009), resulting in a series of spatially constrained 1D resistivity models at the sounding locations (Viezzoli et al. 2008) (Figure 1b). The SkyTEM resistivity models are then assigned to the nearest sampling grid cells by Simple Kriging with a 50 m search radius. The end result is a spatially dense incomplete 3D resistivity grid (Figure 2a). The high spatial density makes it possible to reconstruct the dataset using geostatistical tools, such as pixel based Kriging techniques, a so-called two-point statistical tool, for reconstructing incomplete datasets (Goovaerts 1997). Another approach for reconstruction of incomplete datasets is the method using *DS* presented by Mariethoz & Renard (2010). Since the density of the data points is sufficiently large, the resistivity grid itself can be used as both a TI and soft data variable to stochastically simulate the missing values in the resistivity grid, *i.e.* the gaps in Figure 2a. The MPS dataset reconstruction approach (Figure 2c) is advantageous over the variogram based Kriging estimation (Figure 2b) since it only requires setting up a few parameters. Furthermore, the *DS* approach uses MP information to condition the reconstruction of the dataset. Here, it is important to note that the Kriging method is an estimation method, while the *DS* approach is a simulation method. An estimation method estimates a "best" value, while a simulation method makes a stochastic ensemble of equiprobable guesses. The end result of the *DS* reconstruction approach is an ensemble of stochastic resistivity grids, of which one realization is compared against a corresponding Kriging reconstructed grid in Figure 2b and c. The close ups of Figure 2b and c reveal some key differences in the reconstruction of gaps using Kriging and *DS*. The resistive peak fringing the border of the gap in the westernmost resistive buried valley is smeared into the gap in the Kriging reconstructed grid; see close up in Figure 2b. However, the single *DS* reconstruction presented here does not smear the resistive peak into the gap;



see close up in Figure 2c. The usage of MP information in *DS* allows the possibility that the resistive peak is not part of the

gap.

The uncertainty related to the stochastic resistivity grids is different from the Kriging resistivity grid uncertainty. The standard deviation (STD) related to the Kriging reconstructed grid is closely related to the distance to the nearest data point (Figure 2d), whereas the uncertainty on the stochastic resistivity grids reveals values much more correlated to the patterns of the geophysical information.

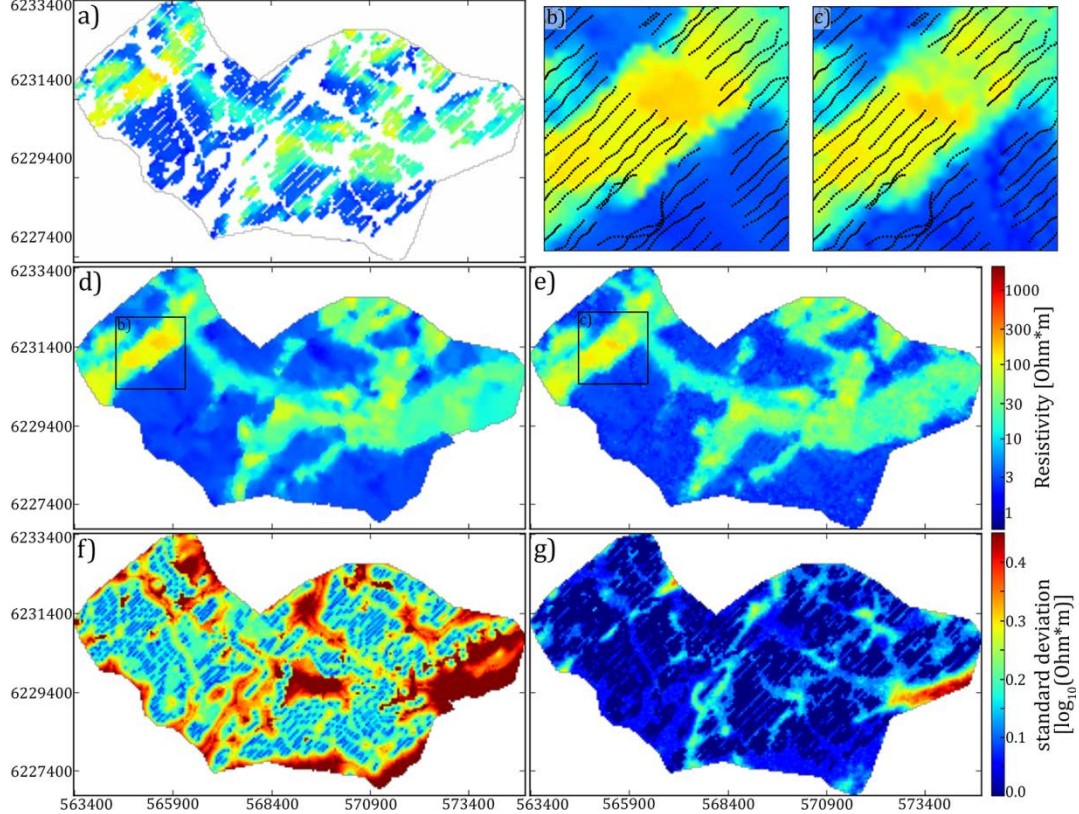


**Figure 2: Comparison of the deterministic Kriging and stochastic DS resistivity grid reconstruction and their corresponding standard deviation. The presented horizontal slice is centered on 20 mbsl. a) shows the incomplete resistivity grid using simple Kriging with a Kriging radius of 50m, b) the reconstructed resistivity grid using Kriging with a close-up of a reconstructed resistive valley, c) a single realization of the reconstructed resistivity grid using DS with the same close-up d) the standard**
**deviation from the reconstructed resistivity grid in b) using Kriging, and e) the standard deviation calculated from an ensemble of 51 stochastic reconstructed resistivity grids using *DS*.**

It is important to note that the resistivity parameter uncertainty has neither been included in the Kriging nor the *DS* reconstruction, enabling the comparison of the STD maps. As an example, a gap present in the homogeneous conductive units with resistivity values between ~2-8 Ωm, has a low STD. According to the TI there is a high probability of finding a
conductive unit in a gap surrounded by only conductive units due to the homogeneity of such conductive units. However,





gaps fringing the border of two contrasting resistivities have large STD values, since information regarding the exact location of the boundary is missing in the TI; *e.g.* the large STD value at the eastern border of the survey area seen in Figure 2e.

In summary, the uncertainty of the *DS* reconstruction provides additional information regarding the reconstructed resistivity patterns over for instance a kriging approach. Also, the MPS reconstruction of the incomplete dataset is less smooth, easier to parameterize, stochastic, and the uncertainty is related to pattern reconstruction and not the distance to the nearest data point.

### 3.3 Hydrostratigraphic modelling setup

The MPS grid reconstruction procedure is used to generate an ensemble of resistivity grids without gaps (Mariethoz and Renard 2010). The reconstructed resistivity grids are used as soft data for constraining the simulation of the hydrostratigraphic models, with the cognitive 3D hydrostratigraphic model used as a TI. The full cognitive geological model contains a total of 42 different geological units (Høyer et al. 2015a), which have been grouped together to form three key hydrostratigraphic categories. The three categories are as follows:

1. *sand & gravel*: Miocene sand, Quaternary meltwater sand and sand till, within and above the Quaternary buried valleys.
2. *glacial clay*: Quaternary clay till and, meltwater clay within and above the buried valleys.
3. *hemipelagic clay*: Hemipelagic, fine grained Paleogene and Oligocene clays.

The simplified cognitive hydrostratigraphic model is used as a TI, and contains the most significant hydrostratigraphic units. Such 3D voxel TIs are usually not readily available, and in most cases 3D TIs are fabricated *ad-hoc*, and are merely conceptual. However, in this case the TI is actually the model we wish to simulate. The justification for this choice of TI lies in that this study is a *proof-of-concept* study, where three different MPS methods are compared against each other. Using a detailed TI containing the desired hydrostratigraphic concepts showcases how well the MPS methods perform in a stochastic hydrostratigraphic modelling workflow with a relevant TI.

The overall workflow can be seen in Figure 3. In detail, the steps are:

1) The SkyTEM resistivity grids are reconstructed using the methodology of Mariethoz & Renard (2010) as described in section 3.2 "Reconstructing incomplete dense geophysical datasets".
2) The ensemble of reconstructed SkyTEM resistivity grids is used as soft data for constraining the three MPS methods:
   a. A reconstructed resistivity grid and the TI are used in the *snesim* framework:



    i. Using histograms created using the Resistivity Atlas approach presented by Barfod *et al.* (2016) (Figure 4c and d) a single reconstructed resistivity grid is translated into a set of probability maps (Figure 5)

    ii. The TI is used for conditioning in conjunction with the probability maps which are used for spatially constraining the *snesim* simulations using the tau model (Journel 2002). The end
    result is a realization of a hydrostratigraphic model

  b. A reconstructed resistivity grid is selected and used in combination with the TI for running *DS*:

    i. The soft data variable (the resistivity grid) is used for both constraining and as the auxiliary variable. The soft data grid is directly available as an auxiliary variable since it geographically overlaps with the categorical TI variable. The combination of the cognitive hydrostratigraphic
    model and auxiliary variable create a bivariate TI

    ii. The bivariate TI is used together with the soft data grid to simulate a realization of the hydrostratigraphic model

  c. A reconstructed resistivity grid is used together with the TI for running *iqsim*:

    i. As with *DS*, the soft data grid is used as an auxiliary variable, and for spatially constraining
    the simulations. The TI and auxiliary variable are combined into a bivariate TI.

    ii. The bivariate TI is used to create a simulation of the hydrostratigraphic model.

Steps 2a-c are repeated N times, once for each reconstructed resistivity grid. In this study N=51. For each of the 51 reconstructed soft data grids three simulations have been run, one simulation per MPS methods: *snesim*, *DS* and *iqsim*, yielding a total of 153 hydrostratigraphic realizations.

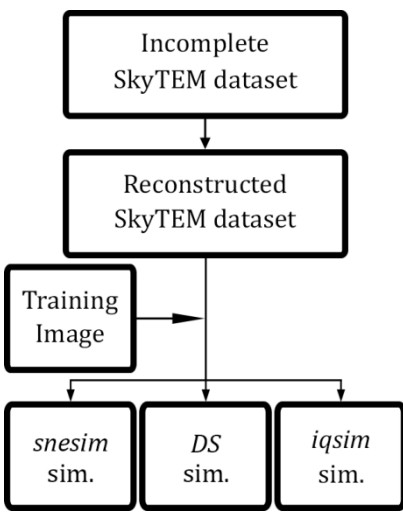

**Figure 3: Workflow diagram showing the stochastic modelling procedure for a single realization. Each simulation is run with snesim, DS and iqsim.**





### 3.4 The hydrostratigraphic-resistivity relationship

Spatially constraining the simulations to the soft data requires information regarding the relationship between

hydrostratigraphic units and, in this case, resistivity values. In *DS* and *iqsim* the information is contained in the bivariate TI, which in this case consists of a categorical and a continuous auxiliary variable. As discussed in section *3.1.2 Direct sampling simulation - DS,* the setup used in this paper avoids using the G$^*$ operator due to the geographically overlapping resistivity and cognitive hydrostratigraphic model grids. This also enables summarizing the hydrostratigraphic-resistivity relationship as a set of histograms (Figure 4a and b). The histograms summarizing the hydrostratigraphic-resistivity relations used in *DS*

and *iqsim* are seen in Figure 4a, and the corresponding summary statistics are found in Table 1. These histograms are created by selecting one of the reconstructed resistivity grids and combining it with the TI. The same relationship is seen in Figure 4b, however, instead of using the *DS* reconstructed resistivity grid the Kriging reconstructed grid is used instead. The main difference between the two sets of histograms are a slightly larger separation of the *sand & gravel* and the *glacial clay* for the Kriging reconstructed grid (Figure 4a and b) (Table 1). For the *DS* reconstructed grid the median values for *sand &*

*gravel* and *glacial clay* histograms are 48 Ωm and 32 Ωm, respectively. While for the Kriging reconstructed grid the median values are 46 Ωm and 27 Ωm, respectively. Furthermore, the Kriging *sand & gravel* histogram is wider with an interquartile range of 38 Ωm, which for the *DS* grid was 31 Ωm.

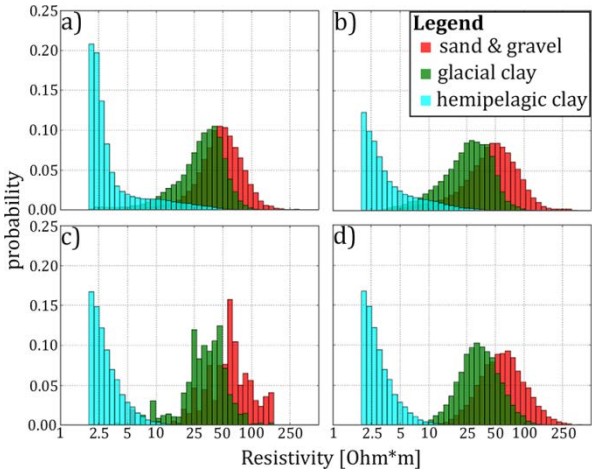

**Figure 4: The hydrostratigraphic-resistivity relation shown as a series of histograms; a) shows the histograms created by**
**categorizing the *DS* reconstructed resistivity grid according to the simplified hydrostratigraphic model created by Høyer et al. (2015a), b) the histograms created by categorizing a resistivity grid which has been reconstructed using Kriging, c) the histograms resulting from the Resistivity Atlas approach presented by Barfod et al. (2016), and d) the Resistivity Atlas histograms have been reproduced based on the summary statistics from c) to create a set of lognormal histograms.**

In the *snesim* framework constraining to the soft data requires a translation of the soft resistivity data into a set of probability
maps, one for each of the hydrostratigraphic units. This is achieved by using prior information regarding the hydrostratigraphic-resistivity relationship. Often this information is difficult to obtain, unless a large number of boreholes are available. If boreholes are readily available the Resistivity Atlas framework (Barfod et al. 2016) can be utilized. The raw



Resistivity Atlas histograms are seen in Figure 4c. Due to the general coarse nature of the histograms the mean and interquartile range from the coarse histograms (Figure 4c) were computed and used to create a set of smooth histograms with identical summary statistics (Figure 4d). By comparison the Resistivity Atlas histograms are quite similar to the Kriging grid histograms (Figure 4b). However, the separation between the *sand & gravel* and *glacial clay* histograms is even larger in the Resistivity Atlas histograms. The respective median values are 59 Ωm and 34 Ωm. The *sand & gravel* histogram also has a quite large spread with an interquartile range of 43 Ωm (Figure 4c and d) (Table 1).

Table 1: The summary statistics table for the histograms in Figure 4. The first section, named DS, shows summary statistics for the three histograms seen in Figure 4a. The second section, named Kriging, shows the summary statistics for the histograms in Figure 4b. The last section, labelled Resistivity Atlas, shows the summary statistics for the Resistivity Atlas histograms Figure 4c and d. All the values presented in the table are resistivities [Ωm].

| | 25th percentile | Median | 75th percentile | IQR |
|---|---|---|---|---|
| **DS** | -- | -- | -- | -- |
| *Sand & gravel* | 34.2 | 47.6 | 65.5 | 31.3 |
| *Glacial clay* | 21.1 | 31.8 | 42.9 | 21.8 |
| *Hemipelagic clay* | 2.2 | 2.6 | 3.7 | 1.5 |
| **Kriging** | -- | -- | -- | -- |
| *Sand & gravel* | 29.7 | 46.4 | 67.9 | 38.2 |
| *Glacial clay* | 17.2 | 26.6 | 38.2 | 21.0 |
| *Hemipelagic clay* | 1.9 | 2.4 | 3.6 | 1.8 |
| **Resistivity Atlas** | -- | -- | -- | -- |
| *Sand & gravel* | 38.4 | 59.2 | 81.4 | 43.0 |
| *Glacial clay* | 24.2 | 33.9 | 46.7 | 22.5 |
| *Hemipelagic clay* | 2.1 | 2.5 | 3.4 | 1.3 |

The *hemipelagic clays* have unique properties. They are aquitards with low hydraulic conductivity and often used as a hydraulically confining no-flow boundary at the bottom of a groundwater model in parts of Denmark. When *hemipelagic clay* is encountered during drilling, the drilling is halted and generally *hemipelagic clay* is sparse in Danish borehole lithology logs. For this reason the Resistivity Atlas based on Transient Electromagnetic data does not provide a lot of information on *hemipelagic clays*. However, the hemipelagic clays are regionally extensive and homogeneous. From wireline resistivity logs in eastern Jutland they are found to be conductive, with median resistivities ranging between 4-7 Ωm. Based on this knowledge the *hemipelagic clay* histograms in Figure 4c and d are created.

The model setup is different for the three MPS methods. When running *DS* and *iqsim* the hydrostratigraphic-resistivity relationship is explicitly given due to the geographically overlapping resistivity grid and hydrostratigraphic TI. Normally the





auxiliary variable has to be created for the given TI using the $G^*$ operator. The full $G^*$ approach has been elaborated in section *3.1.2 Direct sampling simulation - DS,* and requires prior knowledge regarding the hydrostratigraphic-resistivity relationship, much like when creating the probability grid for *snesim*. The *snesim* setup, however, avoids using the $G^*$

operator approach, and in place the Resistivity Atlas histograms (Figure 4c and d) can be used to directly translate the resistivity grid into probability grids (Figure 5).

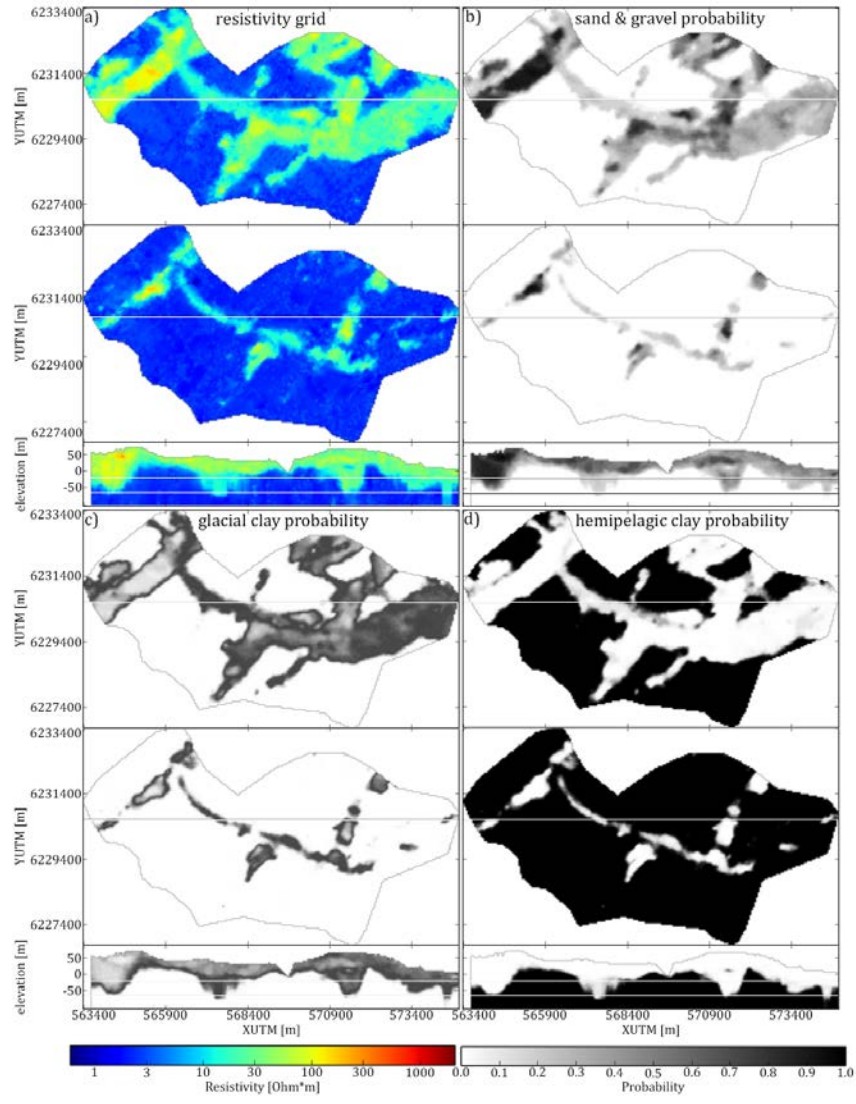

**Figure 5: The SkyTEM soft data grids are translated into three sets of probability grids, one for each lithological category to be simulated; a) shows one of the reconstructed SkyTEM grids, the top frame is a horizontal slice in the 3D grid at 20 mbsl, the**

**second frame is a horizontal slice of the grid layer at 60 mbsl and the bottom frame shows the vertical cross-section intersecting at UTMY coordinate 6230150 m, b) shows the sand & gravel probability grid, c) shows the glacial clay probability grid, and d) shows the hemipelagic clay probability grid. The horizontal slices and the vertical cross sections of b), c) and d) are the same as the ones presented in a).**



### 3.5 The modified Hausdorff Distance – a measure for similarity

Comparing 153 3D models each with 1,187,823 grid cells is not trivial. Visual comparison is used mainly to check if the results are geologically realistic, but a detailed visual comparison would be time consuming and subjective. Therefore, a set of tools are used to compare how similar the simulation results are to each other, and how different they are from the TI.

In this study, a distance measure is used as a measure of similarity between 3D model simulations. The chosen distance measure is the modified Hausdorff Distance (MHD), which is a measure for similarity between two binary images, *i.e.*

dissimilar images have relatively large distances (Figure 6e), while similar images have relatively small distances (Figure 6c). Identical images have a distance of exactly zero (Figure 6b). Firstly, the images we wish to study are summarized as binary images. The pixels for each object we wish to compare are set to one, while the remaining pixels are disregarded as a background variable and set to zero. For a pair of images, ImA and ImB, to be compared, two point sets are defined: $A = \{a_1, a_2, ..., a_{N_a}\}$ and $B = \{b_1, b_2, ..., b_{N_b}\}$, where $a_i, i \in \{1,2,...,N_a\}$ and $b_j, j \in \{1,2,...,N_b\}$ are positional vectors

containing the x, y and z positional coordinates in ImA and ImB for the binary object pixels only, *i.e.* the background variable positions are not included in the point sets. Then the MHD between point sets A and B is defined as follows:

$$MHD(A, B) = max\left(\frac{1}{N_a}\sum_{a \in A} min_{b \in B}\|a - b\|, \frac{1}{N_b}\sum_{b \in B} min_{a \in A}\|b - a\|\right), \qquad (2)$$

where $N_a$ and $N_b$ is the total number of points in point sets A and B, respectively. In the context of this paper, *A* and *B* are our 3D voxel models containing the objects we wish to compare. The Euclidian distances between a given point, *a* from

point set *A*, and all points in point set *B* are computed, and $min(...)$ selects the smallest of these distances. This is repeated for all points in point set *A*, and the average is computed. The same operations are performed for point set B. The maximum value of these two results is then returned.

Dubuisson and Jain (1994) found that the MHD was the best performing distance measure out of 24 different Hausdorff based distance measures in relation to objects matching of images. In order to make the pairwise MHD computation tractable

in 3D, we approximate the MHD between solid geobodies by the MHD between their boundaries. In short, the boundary is the selection of the edges or outlines of the geometric objects, such that the objects are now represented by their outlines instead of the entire objects; see Figure 6b-e. The Roberts Cross Operator (Roberts 1998, Senthilkumaran and Rajesh 2009) is used to select the boundary. Instead of defining the point sets based on the geometric objects themselves, only their outlines are included in the point sets. The point sets containing the outline of the geometric objects are then compared using

the MHD.

A 2D example is presented to illustrate the overall MHD concept in Figure 6. The 3D hydrostratigraphic model and the *DS* modelling results are simplified into 2D horizontal cross-sections from the modelling grid layer centered on 20 mbsl. The initial step is to create a binary version of the hydrostratigraphic simulation model (Figure 6a and b). Here, the *sand & gravel* and *glacial clay* were categorized into a single category, and *hemipelagic clay* was used as the background variable. After



categorization the Roberts cross operator is used to find the boundary of the objects (Figure 6b). The procedure of creating

the binary image and outlines is carried out for all the 51 *DS* simulations. For illustration purposes this example is only

computed for the horizontal cross-section centered on 20 mbsl. The MHD is calculated between each of the 51 horizontal

binary maps, representing the *DS* simulations, and the binary hydrostratigraphic model. The resulting MHDs are then sorted

in ascending order and the binary version of the realizations corresponding to the 1st, 25th, and 51st MHD values are

presented in Figure 6c-e.

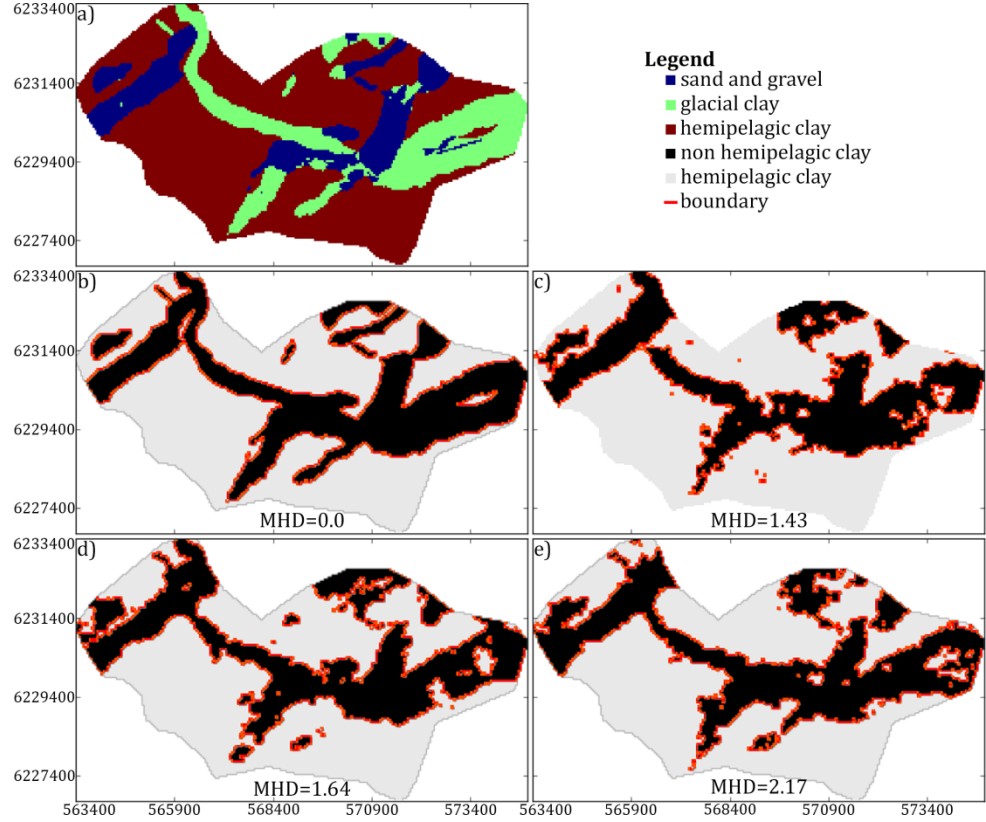

**Figure 6: A 2D example of the binary categorization of the hydrostratigraphic models and example of the Roberts cross operator for edge-tracing; a) a horizontal slice of the hydrostratigraphic model at elevation interval centered on 20 mbsl, b) the result of the binary categorization of the hydrostratigraphic model into two categories: 1) Non hemipelagic clay (black) and 2) hemipelagic clay**
**(white). The boundary of the objects in the binary image is shown in red. c) shows the object and boundary of the objects for the *DS* simulation which has the smallest Modified Hausdorff Distance (MHD), i.e. is the most similar to b). The MHD value is shown in d) shows the *DS* simulation, which has the 25th largest MHD and e) shows the *DS* simulation with the largest MHD and therefore is least similar to the hydrostratigraphic model in b).**

From here on we leave the 2D example, and consider the entire 3D model. In this study the MHD is used as a global distance

measure. A more in-depth analysis of the MHD results is gained by using the "analysis of distance" (ANODI) method (Tan

et al. 2014). The overall goal of ANODI is to provide a framework for comparing realizations from different stochastic MPS

methods. The framework presented by Tan *et al.* (2014) uses the following definition of 'best': "one algorithm A is better




than an algorithm B if the training image statistics are reproduced better while at the same time the space of uncertainty (the variability between realizations) is larger". In the particular MPS setup used in this study, the TI is a relevant cognitive

hydrostratigraphic model and geographically overlaps with the hydrostratigraphic MPS realization grids. Hence the MPS realizations should portray similarity to the cognitive model. In this study a further complexity to the definition of best is added. An algorithm with a large space of uncertainty is not necessarily better, if the resulting models do not reflect the underlying datasets.

The initial step is to create a matrix containing all MHD values between all 153 realizations, and between the individual

realizations and the cognitive model. It is similar to a covariance matrix, but instead of containing covariance values, it contains MHD values. The usage of bold letters refers to a matrix. The full $\boldsymbol{MHD}$ is defined as follows:

$$\boldsymbol{MHD_{i,j}} = MHD\big(real_i, real_j\big), \ where \begin{cases} i = 1, \dots, (N_{reals} + 1) \\ j = 1, \dots, (N_{reals} + 1) \end{cases} \tag{3}$$

Where $real_i$ and $real_j$ denotes the individual hydrostratigraphic realizations, $N_{reals}$ is the total number of realizations, in this case $N_{reals} = 153$, and last row and column of $\boldsymbol{MHD}$ contains the distances between the realizations and the cognitive

model, *i.e.* $real_{N_{reals}+1}$ represents the cognitive model. One $\boldsymbol{MHD}$ matrix is created for all three MPS methods. For each of the three MPS methods, the $\boldsymbol{MHD}$ can be evaluated by itself by calculating the MHD variability, $MHD_{var}$:

$$MHD_{var} = \frac{1}{(N_{reals}/3)^2} \sum_{i=MPS_{start,i}}^{MPS_{end,i}} \sum_{j=MPS_{start,j}}^{MPS_{end,j}} \big(\boldsymbol{MHD_{i,j}}\big) \tag{4}$$

where $N_{reals}$ is the size of $\boldsymbol{MHD}$, in this study $N_{reals} = 3 * 51 = 153$, $MPS_{start,i}$ and $MPS_{end,i}$ are the start and end indexes for the entries related to the given MPS method, $\boldsymbol{MHD_{i,j}}$ is $MHD\big(real_i, real_j\big)$. Note that the distances between the

individual realizations and the cognitive model are not included in the $MHD_{var}$. The $MHD_{var}$ equates to computing the average of the MHD values between the realizations of a single MPS method. The larger the $MHD_{var}$ the more dissimilar the simulation results, meaning they portray a large set of possible hydrostratigraphic architectures. Using equation (4) it is also possible to compute the distances between the realizations of different MPS methods, *e.g.* the average MHD between *snesim* and *DS*.

The other evaluation measure, which can be calculated from $\boldsymbol{MHD,}$ is the distance between the realizations and the cognitive hydrostratigraphic model, or TI, which is summarized by the $MHD_{cog}$, which is computed as follows:

$$MHD_{cog} = \frac{1}{N_{reals}/3} \sum_{i=MPS_{start,i}}^{MPS_{end,i}} \big(\boldsymbol{MHD_{i,N_{real}+1}}\big) \tag{5}$$

where, again, $N_{reals} = 153$, $MPS_{start,i}$ and $MPS_{end,i}$ are the start and end indexes for the entries related to the given MPS method and $\boldsymbol{MHD_{i,N_{real}+1}}$ is the $MHD(real_i, cog.model)$. The $MHD_{cog}$ is the average MHD between each individual

realization and the cognitive hydrostratigraphic model. The larger the average MHD the more dissimilar the



hydrostratigraphic realizations are from the cognitive hydrostratigraphic model. The reason we wish to compare the distance to the cognitive model, is that the cognitive model, geographically overlaps with the hydrostratigraphic MPS realizations.

It is also possible to evaluate the **MHD** using dimensional reduction techniques. Such techniques help us view the high dimensional **MHD** in a 2D and/or 3D map. Such a plot gives us a visual representation of the most significant structures of the **MHD**. For dimensional reduction we use a variation of so-called Stochastic Neighbor Embedding (SNE) (Hinton and Roweis 2002). The technique is called t-distributed Stochastic Neighbor Embedding, or t-SNE (Maaten and Hinton 2008). The t-SNE method is advantageous over other SNE techniques, since it is easier to optimize and produces better visualizations. The idea is to visualize the level of similarity of individual entries, or distances in the **MHD**. The overall goal is to place each MHD value as a point in a 2D space where the relative distances between the point values reflect the degree of similarity. Similar points are close to each other, while dissimilar points are far from each other. This is achieved by t-SNE.

### 3.6 Distance to boreholes

In reservoir modelling boreholes are considered to be hard information, due to their overall high quality. However, in many surveys related to groundwater modelling, boreholes cannot be considered as reliable hard data due to variable quality. Such as seen in Barfod *et al.* (2016) and He *et al.* (2014) where boreholes were divided into quality groups. Therefore the simulations are run without constraining against boreholes, and then the realizations are compared against the boreholes as an independent measure of geological realism. A method for comparing similarity between the simulated hydrostratigraphic models and the boreholes was developed. The method does not use the MHD, which has previously been used for measuring distances. Instead the simple Euclidean distance is used to measure the average distance between each individual hydrostratigraphic realization and the borehole dataset. The first step is to sort the borehole lithology logs according to the respective hydrostratigraphic units, to create a hydrostratigraphic log; see left half of Figure 7. Once this has been carried out three sets of binary and regularized logs are created from the hydrostratigraphic log; see right half of Figure 7. For each sampling grid interval, the presence of the given hydrostratigraphic category, say sand & gravel, is saved in the binary log. The end result is a log which states whether or not sand & gravel is present within the given sampling grid interval; active if present and inactive if not present. Three such binary logs are created, one for each of the hydrostratigraphic categories, i.e. sand & gravel, glacial clay and hemipelagic clay (Figure 7). A binary log grid is created by simply assigning the binary active values to the grid cell in which they are present. The Average Euclidian Borehole Distance, AEBD, between the binary logs and a given realization, $real$ , for a given hydrostratigraphic category, $hydro.\,cat_j$ where $j \in \{1, \ldots, N_{hydro.\,cats}\}$, is calculated as follows:

$$AEBD\big(hydro.\,cat_j\big) = \frac{1}{N_{active}} \sum_{i=1}^{N_{active}} \min \big(\|binlog_i - real(hydro.\,unit_j)\|\big) \tag{6}$$



where $binlog_i$ is i$^{th}$ cell in the binary log grid, $N_{active}$ is the number of active cells in the binary log grid, ED is the Euclidian

Distance, and $real(hydro.\ unit_j)$ is the binary realization grid containing only the j$^{th}$ hydrostratigraphic unit, where in this

case $N_{hydro.\ cats} = 3$.

The end result is three arrays, one for each hydrostratigraphic unit, each containing one average distance per realization for

540   the given MPS method. The distance arrays for each individual MPS method can then be compared to the distance arrays of

the other MPS methods.

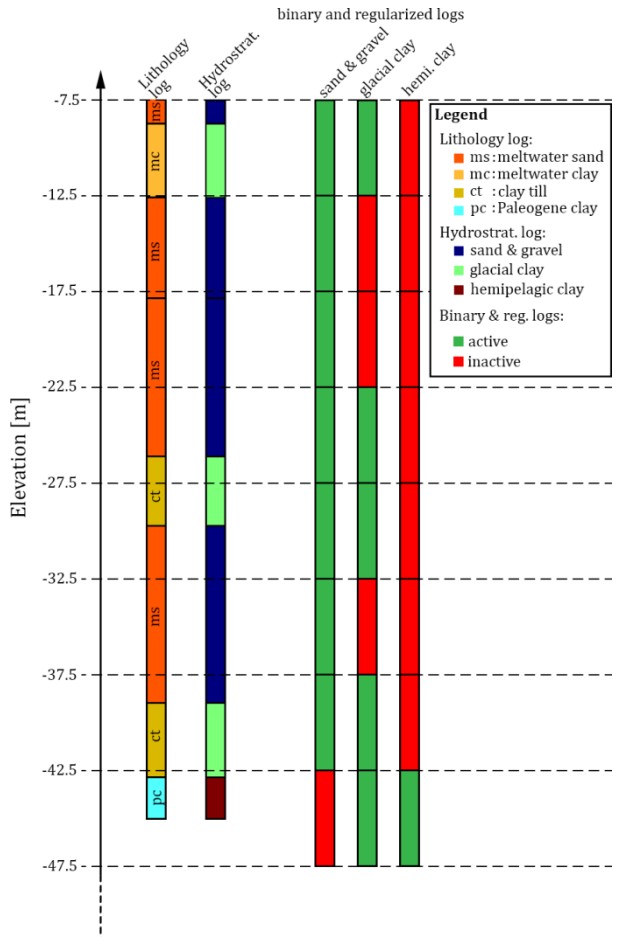

**Figure 7: An example of how a single lithology log is categorized and sorted for the purpose of calculating the borehole distance.**
**The first step is to translate the raw lithology log into a hydrostratigraphic log, which is achieved by categorizing the multiple**
545   **lithological categories into a subset of three hydrostratigraphic categories corresponding to the target categories we wish to model.**
**Note that some categories do not fit into the overall hydrostratigraphic categories and are therefore not translated, e.g. the**
**meltwater silt category in this example. The final step is then to assign the hydrostratigraphic logs to the regularized sampling grid**
**and create one binary log for each of the three target modelling categories. This is done by simply asking whether or not the given**
**hydrostratigraphic category is present (True) or not (False) for the given sampling grid interval.**



## 4 Results

The hydrostratigraphic simulation results include 153 3D hydrostratigraphic realizations, each containing 1,187,823 grid cells. The models can be subdivided into 51 *snesim* realizations, 51 *DS* realizations and 51 *iqsim* realizations. A visual presentation of the hydrostratigraphic model or TI, and two realizations for each of the three different MPS methods is seen in Figure 8. The cognitive hydrostratigraphic model (Figure 8a) shows clear-cut and smooth buried valley architecture with almost no unrealistic short scale variability. Comparing the cognitive hydrostratigraphic model to the stochastic MPS hydrostratigraphic models reveals the more erratic nature of both *snesim* and *DS*, *i.e.* both MPS methods yield models containing unrealistic short scale variability (Figure 8b and c).

Overall *snesim* (Figure 8b) and *DS* (Figure 8c) realizations are similar in nature. In the example provided, Figure 8c, the West-Northwest - East-Southeast trending *glacial clay* valley (see box in Figure 8a) is uninterrupted in one realization, but intersected by *hemipelagic clay* in the other realization. In 47 of the 51 *snesim* realizations, the *glacial clay* valley is uninterrupted, in the remaining 4 realizations the valley is intersected by *hemipelagic clay*. The presented soft data grid in Figure 5d shows a small probability of approximately 10% for *hemipelagic clay* at the position of the valley gap. The 4 realizations which yielded an interrupted *glacial clay* valley amount to 8% of the 51 realizations, which is close to the probability found in the probability grids. The *DS* realizations shows valley architecture with less resemblance to the soft data, *i.e.* the valleys are not conditioned in accordance to the soft data grids. In 11 of the 51 simulation results the valley is intersected by *hemipelagic clay*, amounting to 22% of the 51 realizations.

The *iqsim* results are the most similar to the cognitive hydrostratigraphic model with regards to unrealistic short scale variability, which is generally non-existent. Generally, realizations will reflect the TI, and unrealistic short scale variability is only introduced if present in the TI. This is due to the nature of *iqsim* which is not a pixel based algorithm, like *snesim* and *DS*. Instead, i*qsim* cuts the TI into patches and then reassembles the patches, which means that noise patterns which are smaller than the patch size cannot be fabricated, unless actually present in the TI. The *iqsim* realizations show smooth and clear-cut valley architecture. The main issue with the *iqsim* realizations is that artifacts are introduced near the surface of the model, evident if the vertical *iqsim* cross-sections (Figure 8d) are compared to the remaining vertical cross-sections of the TI, *snesim* and *DS* (Figure 8a-c). This is neither reflected in the resistivity grid (Figure 5) nor in the TI (Figure 8a). Close to terrain hydrostratigraphic layers consist of either *glacial clays* or *sand & gravel*, and conductive *hemipelagic clays* are not evident. Since the soft data does not support the presence of the *hemipelagic clays* in the upper part of the hydrostratigraphic model, the soft data can be concluded to being improperly constrained with this specific setup. Another observation is that in 43 out of the 51 realizations, amounting to 84%, the referenced *glacial clay* valley is intersected by *hemipelagic clay*.



**Figure 8: The hydrostratigraphic MPS realizations are presented as horizontal slices centered on 20 mbsl and vertical cross-sections intersecting at UTMY 6230150m; a) shows the cognitive hydrostratigraphic model, with a West-Northwest - East-Southeast trending *glacial clay* valley marked by a box, b) shows two *snesim* realizations, c) shows two *DS* realizations, and d) shows two *iqsim* realizations.**



An advantage of the *iqsim* implementation used (Hoffimann et al. 2017) is the favorable computation time. On an Intel® HD Graphics Skylake ULT GT2 GPU of a Dell XPS 13 laptop, *iqsim* runs with an average simulation time of 10-12 min per realization with the attempted setup. On a different laptop running a 64bit Windows system, with 8GB RAM, an SSD hard disk, with an Intel core i7-3520 M CPU at 2.9 GHz, the computation times for *snesim* were on average between ½-1 h. Since the *DS* computation times were significantly larger at 6 h 15 min per realization, the *DS* simulations were run on a 64 bit

Windows server with 64 AMD Opteron processor 5376 at 2.3 GHz each, with a total of 128 GB RAM and a SSD hard disk. The implementation of *DS* used in this paper is called *DeeSse* (Straubhaar 2011) and is easy to script and run in parallel on a server or computer cluster. The total time required for 51 simulations running in parallel was approximately 32h, without enabling parallelization which is available in *DeeSse*. One *DS* simulation took between 6-7 h. For more detailed information see Table 2 which summarizes the computation time for the three MPS methods.

**Table 2: A table presenting the average computation times per realization for each of the three MPS methods and the approximated computation times needed for running 51 realizations with the setup used in this study. *indicates that the given realizations were run in parallel on a server; other realizations were generated on a personal laptop.**

|                                    | *snesim*    | *DS*     | *iqsim*     |
| ---------------------------------- | ----------- | -------- | ----------- |
| *Comp. times pr. realization*      | ½-1 h       | 6-7 h*   | 10-12 min   |
| *Approx. comp. times  51 realizations* | 38 h 15 min | 32 h*    | 9 h 21 min  |

### 4.1 Modified Hausdorff distance results

The full MHD matrix is presented in Figure 9a and b. Using eq. (4) and (5) the MHD is summarized in Table 3, without the

usage of dimensional reduction techniques. The method with the largest variability, *i.e.* least similar hydrostratigraphic realizations, is *iqsim* with a $MHD_{var}$ of 1.79. The *snesim* and *DS* models show generally lower $MHD_{var}$ values of 0.48 and 0.78, respectively. This means that the *iqsim* results span the largest set of possible models. The *iqsim* realizations also have the smallest average MHD between the individual realizations and the cognitive hydrostratigraphic model, with a $MHD_{cog}$ of 2.65, meaning on average *iqsim* realizations resemble the cognitive hydrostratigraphic model the most. The *snesim* and *DS*

$MHD_{cog}$ are 3.01 and 2.80, respectively. On average the *DS* realizations are more similar to the cognitive hydrostratigraphic model in comparison to the *snesim* realizations, while both are more dissimilar than the *iqsim* realizations. The two MPS methods which had the smallest distances, and therefore were most similar, were *snesim* and *DS* with an inter MHD distance of 1.05. The distance between *DS* and *iqsim* was larger, with a value of 2.19, while the largest inter MHD distance was between *snesim* and *iqsim* with a MHD value 2.37.






**Table 3: Summary of the Modified Hausdorff Distance (MHD) matrix portraying the MHD results.** $MHD_{var}$ **represents the variability of the given MPS method or between the different MPS methods, e.g. the value in the** $MHD_{var}$ **column and the** $snesim \rightarrow DS$ **row represents the average distance between the snesim and DS realizations. The MHDcog is the average distance from the simulation results to the cognitive model.**

|  | $MHD_{var}$ | $MHD_{cog}$ |
|---|---|---|
| *snesim* | 0.48 | 3.01 |
| *DS* | 0.78 | 2.80 |
| *iqsim* | 1.79 | 2.65 |
| $snesim \rightarrow DS$ | 1.05 | --- |
| $snesim \rightarrow iqsim$ | 2.37 | --- |
| $DS \rightarrow iqsim$ | 2.19 | --- |

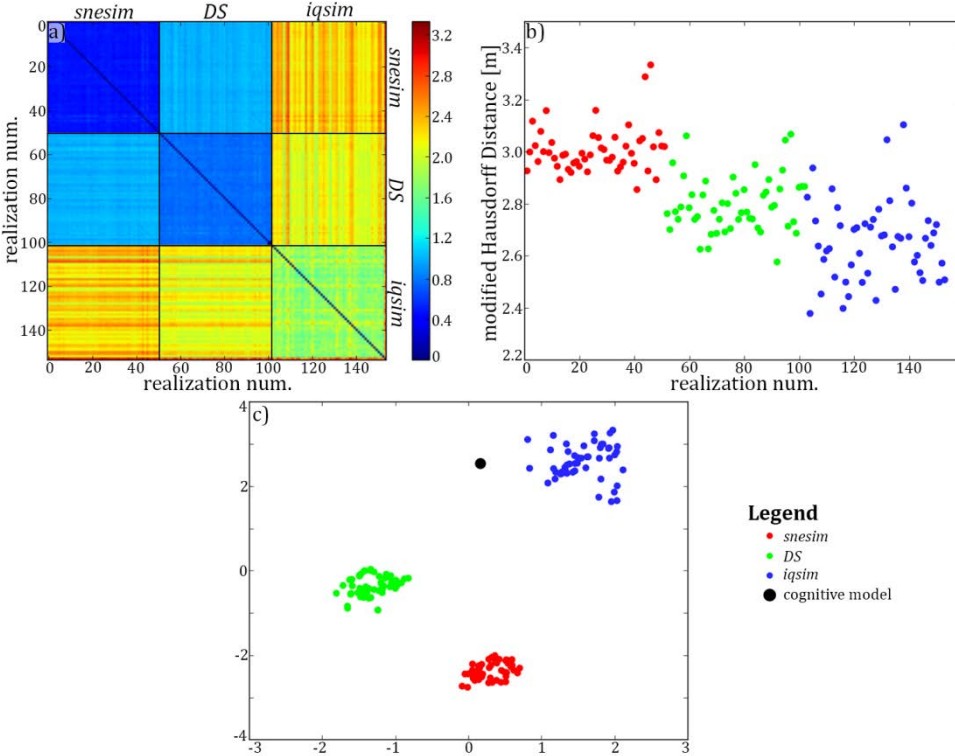

**Figure 9: The MHD results presented without and with dimensional reduction. a) the full Modified Hausdorff Distance (MHD) matrix showing the distances between individual realizations, and between individual realizations and the cognitive hydrostratigraphic model. The last column and row of the distance matrix of a) represent the distances between the realizations and the cognitive model. b) shows a scatter plot of these distances between the realizations and the cognitive model, revealing greater detail than can be seen with naked eye from the MHD matrix itself. c) shows the 2D t-SNE plot of the MHD matrix.**





The **MHD** can also be evaluated by applying the aforementioned t-SNE method. Here, each realization is visually represented as a point in 2D space. Similar values, with small MHD values, are closely spaced, while dissimilar values, with

large MHD values, are separated from each other. Firstly, the t-SNE results show *snesim* and *DS* point clouds which are closer to each other relative to the *iqsim* point cloud (Figure 9c). This means that they are similar in nature; as reflected in Table 3. The *iqsim* point cloud is isolated in the 2D space since the *iqsim* realizations are significantly different from the *snesim* and *DS* results. Furthermore, the *iqsim* point cloud is also the largest, which reflects the larger dissimilarity of the output realizations. On average, the *iqsim* point cloud is closer to the cognitive model, which is also reflected in Figure 9b

and Table 3.

### 4.2 Borehole validation results

The final comparison of the MPS methods regards the average Euclidean distance between the simulation results and the regularized binary hydrostratigraphic logs. The sorted average distances between each individual simulation and the boreholes are seen in Figure 10.

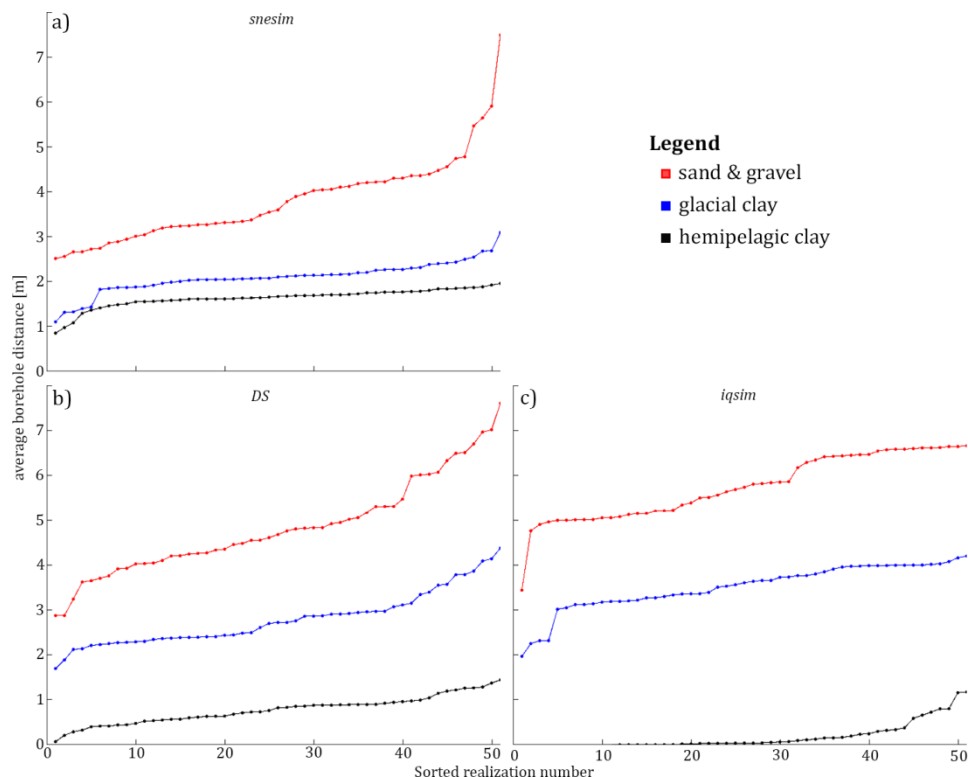


**Figure 10: The borehole distance results are presented for each of the three MPS methods: *snesim*, *DS*, and *iqsim;* a) shows the *snesim* borehole distance results for the three hydrostratigraphic units, b) shows the *DS* borehole distance results for the three hydrostratigraphic units, and c) shows the *iqsim* borehole distance results for the three hydrostratigraphic units**



The average distance between the simulated hydrostratigraphic models and the boreholes are presented according to the three

key hydrostratigraphic units. The average distance between *sand & gravel* units in the hydrostratigraphic realizations and *sand & gravel* units in the boreholes seems to be the largest for the modelling results of all three MPS methods (Figure 10), *i.e.* the red curve is always on top. The average values of the individual curves in Figure 10 are computed and presented in Table 4. The *sand & gravel* average in Table 4 reflects the large distances between resistive *sand & gravel* units in the realizations and the hydrostratigraphic logs. By comparing the individual frames of Figure 10 it is seen that the average value

for the hydrostratigraphic models created using *iqsim* have a higher average distance. The *iqsim* averages for *sand & gravel* is centered on 5.8 m, while for *snesim* and *DS* it is centered on 3.8 m and 4.9 m, respectively. The *iqsim* average distance to *glacial clay* is centered on a relatively large value of 3.5 m, as opposed to 2.1 m and 2.8 m for *snesim* and *DS*, respectively. The *hemipelagic clay* units show a different pattern where *iqsim* has the lowest average distance of 0.2 m, while the *snesim* and *DS* distances are 1.6 m and 0.8 m, respectively.

**Table 4: The borehole distance results are summarized in this table. The borehole distances are the 3D Euclidean distances calculated using the concept presented in section 3.6 "Distance to Boreholes". The presented distance values are the averages of the curves shown in Figure 10, one average for each of the individual hydrostratigraphic units for each of the presented methods: *snesim*, *DS* and *iqsim* realizations.**

|  | sand & gravel [m] | glacial clay [m] | hemipelagic clay [m] |
|---|---|---|---|
| *snesim* | 3.8 | 2.1 | 1.6 |
| *DS* | 4.9 | 2.8 | 0.8 |
| *iqsim* | 5.8 | 3.5 | 0.2 |

**4.3 Hydrostratigraphic modelling of new surveys**

In areas of groundwater interest the initial step is to collect different types of data relevant to the hydrological properties of the subsurface. Among these data are dense geophysical datasets, *e.g.* SkyTEM, which can be collected quickly and usually cover a significant part of the survey area. The different datasets are processed and modeled, and used in conjunction with the borehole lithology logs to create a single geological and/or hydrostratigraphic model. This model is only one version of the subsurface, encasing only part of the complexity related to the given hydrological system. We present a practical

example of stochastic simulation of hydrostratigraphic models. The end result is multiple hydrostratigraphic realizations, covering a larger span of possible models. Using the cognitive hydrostratigraphic model from area A as a TI, another hydrostratigraphic model from survey area B is simulated, using only the geophysical data in area B for spatial constraining. An important assumption is that the geological settings of area A and B are similar, since the hydrostratigraphic information is shared through the TI from area A. Furthermore the hydrostratigraphic-resistivity relationship needs to be stationary so

that it can be assumed that the hydrostratigraphic-resistivity relationships are statistically comparable.



The example presented in this study is synthesized from the Kasted dataset. The dataset is divided in two along the UTMX coordinate 569025m (Figure 11a). The left half of the cognitive hydrostratigraphic model is then used as a TI to simulate the right half of the model. The reconstructed resistivity grid is also cut in half Figure 11b). The left half of the resistivity grid is used as an auxiliary variable describing the hydrostratigraphic-resistivity relationship, as seen in Figure 4a, while the right

half is used for spatially constraining the simulation. In this example 10 stochastic hydrostratigraphic realizations are created using *DS*. A single hydrostratigraphic realization is seen in Figure 11c, while the mode of the hydrostratigraphic model ensemble is seen in Figure 11d.

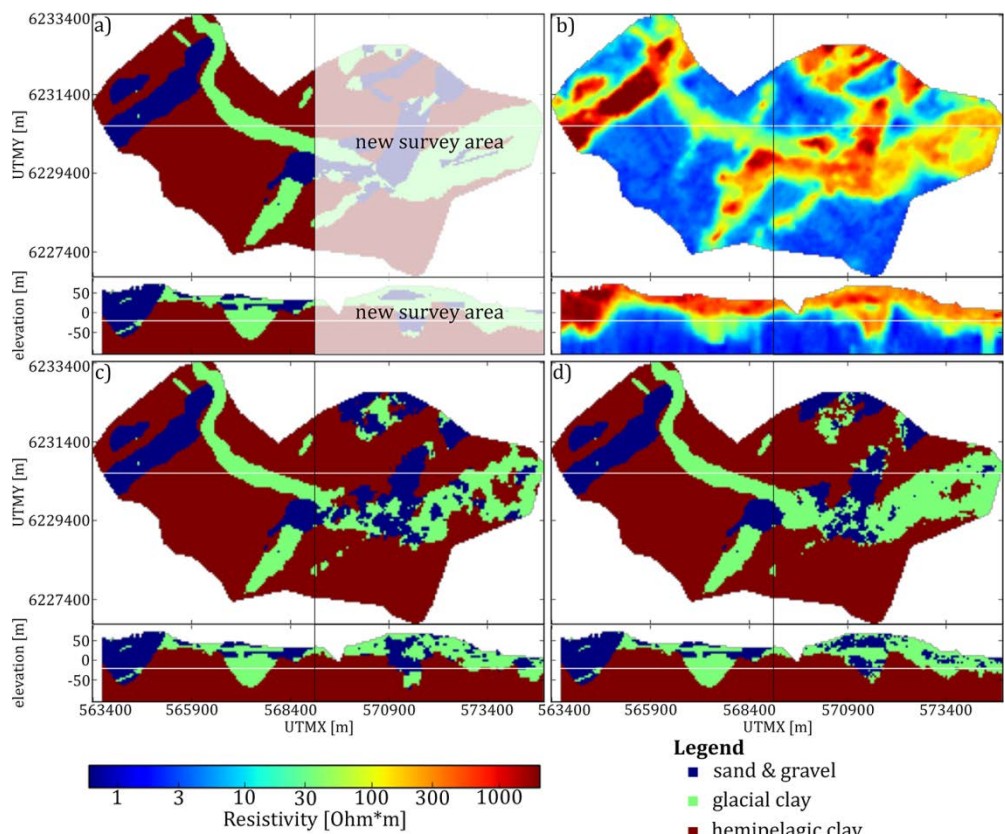

**Figure 11: An overview of the setup for simulating new survey areas and the hydrostratigraphic modelling results using the**
**Kasted dataset. The presented horizontal slices are centered on 20 mbsl, and the vertical cross-section intersects at UTMY 6230150m. a) the cognitive hydrostratigraphic model is cut in half to simulate having two survey areas, one area with a cognitive hydrostratigraphic model (training image) available and the other without. The white area represents the new survey we wish to simulate. b) the horizontal slices and vertical cross-sections of the soft data used to simulate the new area. The left half is the auxiliary variable, while the right half constrains the simulation of the new survey area. c) a single hydrostratigraphic realization.**
**The left half is exactly the same as the cognitive model, see a), while the right half is simulated using *DS*. d) the mode of an ensemble of 10 hydrostratigraphic model realizations. Again, the left haft is the same as the training image and the right half shows the ensemble mode of 10 realizations.**




The simulation results show that one hydrostratigraphic realization represents the overall architecture of the resistivity grid; compare Figure 11c and b. Comparing the single hydrostratigraphic realization (Figure 11c) to the original cognitive model (Figure 11a) reveals that one realization largely reflects the variability in the soft data grid. The mode of the model ensemble on the other hand (Figure 11d) has a closer resemblance to the cognitive hydrostratigraphic model; compare Figure 11a and d. This means that the individual realizations do on average resemble the original cognitive model. The end goal is not to create a set of hydrostratigraphic models which match the cognitive hydrostratigraphic model. The goal is to create a suite of realistic hydrostratigraphic models.

Generally, unrealistic short scale variability is introduced in both the single hydrostratigraphic realization as well as in the ensemble mode model, but is generally not present in either the TI or the resistivity grid. These short-scale variation patterns are artifacts from the *DS* method itself and can be removed by using post-processing tools (Pyrcz and Deutsch 2014). Such tools are generally run on the realizations to remove artifacts that were introduced due to the algorithms. Since post-processing is a separate step, it was not used for any of the simulations in this study, making all the simulation results comparable to each other.

## 5 Discussion

The *snesim* setup is different from the *DS* and *iqsim* setups. The *snesim* setup differs in the usage of the probability framework, and in the choice of the implicit Resistivity Atlas histograms (Barfod et al. 2016). The implicit histograms (Figure 4d) are used to directly translate the resistivity grids into probability grids. This illustrates the utility of the Resistivity Atlas framework in relation to geostatistical modelling. The *DS* and *iqsim* frameworks would normally, in real-world cases, require the usage of a $G^*$ operator since no auxiliary variable exists which geographically overlaps with a conceptual TI. As explained in section *3.1.2 Direct sampling simulation - DS,* applying a realistic $G^*$ operator requires several steps and can be a complicated affair. In this study, however, the TI was an actual cognitive geological model of Kasted study area, meaning a resistivity grid which geographically overlaps with the TI exists. Using the SkyTEM resistivity grid as an auxiliary variable resulted in the application of different resistivity-hydrostratigraphic relationships in the *DS* and *iqsim* approach – compare the explicit histograms used in *DS* and *iqsim*, Figure 4a, with the implicit Resistivity Atlas histograms used in *snesim*, Figure 4d, or see Table 1. Even though there are some differences in the setups of the different *MPS* algorithms, the *snesim* and *DS* realizations are still similar in nature; compare Figure 8b and c. The differences mentioned here are mainly due to the differences of the implementation of the algorithms.

The MHD results revealed some interesting trends between the MPS realizations. The $MHD_{var}$ for *snesim*, *DS* and *iqsim* were 0.48, 0.78, and 1.79, respectively. The low *snesim* $MHD_{var}$, is related to the soft data conditioning, which is dependent on the choice of histograms for translating the resistivity grids into probability grids. For this translation, as mentioned, the implicit Resistivity Atlas histograms were used. Overall, the implicit histograms show a larger separation between the



*glacial clay* and *sand & gravel* histograms compared to the explicit histograms; compare Figure 4a and d. This results in less
ambiguity in the transition from *glacial clay* to *sand & gravel* in the probability grids, yielding a smaller subset of possible models. This also results in *snesim* realizations which closely resemble the soft data variable, compared to *DS* and *iqsim*. The borehole distance results are also influenced by choosing the implicit histograms. Generally, the *snesim* realizations show the smallest borehole distances with respect to *glacial clay* and *sand & gravel* units, while the corresponding *hemipelagic clay* distances are the largest. The increased separation of the *glacial clay* and *sand & gravel* histograms seem to improve the
*snesim* borehole distances to these units, while *hemipelagic clay* seems to be underestimated.

It can be concluded that *snesim* and *DS* yield similar realizations, portrayed by the relatively small MHD values between *snesim* and *DS*. This is reflected in the t-SNE plot (Figure 9c), where the *iqsim* point cloud is isolated from the *snesim* and *DS* point clouds, and is closer to the cognitive model. The isolation of the *iqsim* point cloud agrees with a lack of short-scale variability in the *iqsim* realizations compared to snesim and DS. However, the abundance of *hemipelagic clay* close to terrain
is clear and undesired in iqsim realizations; see the vertical cross-sections of the models (Figure 8). Evidence of abundant near surface *hemipelagic clay* is also found in the borehole distance results. The borehole distances of the *iqsim* realizations revealed exceedingly small *hemipelagic clay* distances, with an average of 0.2m. In comparison, *snesim* and *DS* had *hemipelagic clay* borehole distance averages of 0.8m and 1.6m, respectively. This shows that *iqsim* produces realizations where *hemipelagic clay* units are, on average, closer to the borehole hydrostratigraphic logs. However, it is important to also
notice the relatively large *iqsim* borehole distances for *glacial clay* and *sand & gravel* units. This indicates that the ample near surface *hemipelagic clay*, decreases the *hemipelagic clay* borehole distances, while increasing the *glacial clay* and *sand & gravel* borehole distances (Figure 10) (Table 4).

Unrealistic short scale variability is found throughout the *snesim* and *DS* realizations. This is an artifact introduced by the algorithms, and do not reflect the underlying datasets, *i.e.* the soft data or TI. As Linde *et al.* (2015) discuss fine-scale
patterns are present in the real-world hydrostratigraphic subsurface, but are not present in geophysical models. Two of the three presented stochastic MPS methods introduce fine-scale variations in the form of short scale variability to the overall hydrostratigraphic architecture, with the overall architecture resembling the underlying datasets. This adds complexity to the realizations and the resulting equiprobable hydrostratigraphic models span a larger subset of possible models. The question, however, is whether this short-scale variability is similar to the real-world short-scale variability missing from our
geophysical data, which is difficult to answer. The importance of short-scale variability also depends on the type of prediction for which the hydrostratigraphic model is to be used.

An important difference in the *iqsim* realizations, compared to *snesim* and *DS,* is the lack of fine-scale variability. The *MHD* results reveal that the *iqsim* realizations were the most similar to the cognitive model, and that they were different from the *snesim* and *DS* realizations. It can be concluded that since *iqsim* realizations do not contain fine-scale variability, the
$MHD_{cog}$ is smaller and the most significant features of the cognitive model are reproduced. It is noted that the MHDs are not



sensitive towards the hydrostratigraphically unrealistic placement of *hemipelagic clay* at the surface in the *iqsim* realizations. The presented MHD results reveal that *iqsim* performs "best", according to the definition of "best" introduced in section "3.5 The modified Hausdorff Distance – a measure for similarity". The TI statistics are reproduced better and the space of uncertainty is large. However, the *iqsim* realizations do not reflect all complexities of the underlying datasets, which is also

reflected by the poorer borehole distance results for *glacial clay* and *sand & gravel* units.

The *snesim* and *DS* realizations portray some differences, which are related to the choice of the implicit Resistivity Atlas histograms for translating the resistivity grid. This can help us understand some of the basic differences in the information provided by the implicit Resistivity Atlas histograms and the explicit auxiliary variable. In *DS* probable hydrostratigraphic units are not conditioned properly. An example of this is the aforementioned West-Northwest - East-Southeast trending

*glacial clay* valley (see Figure 8a), which is uninterrupted in 78% of the *DS* realizations. The same valley is clearly represented in the resistivity grid (Figure 5a) and in the cognitive model (Figure 8a). However, the explicit auxiliary variable histograms show increased overlapping resistivity values for the *glacial clay* and *sand & gravel* histograms (Figure 4a,d). The auxiliary variable histograms (Figure 4a) reveals approximately equal probability of *glacial clay* and *sand & gravel* resistivities lying close to 40-45 Ωm. The histogram also shows that *hemipelagic clay* has a resistive tail, resulting in a small

probability for *hemipelagic clay* in the areas of intermediate resistivity values of 10-50 Ωm. The Resistivity Atlas histograms (Figure 4d), on the other hand, favor the glacial clay in the 40-45 Ωm range, with a 0% probability for *hemipelagic clay*. The *snesim* realizations show an uninterrupted *glacial clay* valley in ~90% of the realizations in the horizontal cross-section centered on 20 mbsl. The probability grid for *snesim* reveals a ~75% probability for *glacial clay* at the location of the West-Northwest - East-Southeast trending *glacial clay* valley at 20 mbsl (Figure 5c). At 20 mbsl the *sand & gravel* probability

~15-20% while *hemipelagic clay* has a low probability of ~0-5% (Figure 5b,d). The underlying hydrostratigraphic-petrophysical relationship which holds information on how to condition the simulations to the soft data, is important to the *MPS* modelling results, especially when extensive and spatially dense geophysical datasets are available.

The presented practical example, showing the simulation of a new survey, has the two aforementioned requirements: 1) the geological environments of the two areas need to be similar and 2) the statistical hydrostratigraphic-petrophysical relations

also need to be similar. Since the Resistivity Atlas histograms are created using only local data, *i.e.* boreholes and SkyTEM resistivity models, they represent the local relationship. Since stationarity in the hydrostratigraphic-petrophysical relations is not guaranteed (Barfod et al. 2016), it is necessary to check for stationarity, which is possible within the Resistivity Atlas framework. Here, histograms can be created for each area and compared. If statistically similar, stationarity can be inferred for the hydrostratigraphic-petrophysical relations.

The hydrogeophysical dataset is processed and modeled for the purpose of creating a petrophysical model. The practical example has the advantage of an explicit implementation of hydrostratigraphic-resistivity relationship by using the resistivity grid as an auxiliary variable. The relationship is indirectly modeled during the cognitive modelling process. During modeling



the geoscientist makes qualitative decisions regarding the relations between geophysical and borehole data. Experienced geoscientists have a general understanding of how geophysical data reflect the geological features. That knowledge is used to
create geologically realistic hydrostratigraphic models, resembling both the geophysical and the sparse borehole data. The explicit hydrostratigraphic-petrophysical relationship is described in detail, and can be extracted from the collocated hydrostratigraphic and petrophysical grids; as presented in Figure 4a.

**6 Conclusion**

The three MPS methods *snesim*, *DS* and *iqsim* are used for stochastic hydrostratigraphic modelling. The modelling results
are compared in an elaborate framework of comparing the modelling results visually, mathematically and against boreholes. Each individual MPS method has its own set of advantages/disadvantages which are covered in this study. Overall the *DS* method had the highest computation times. An average *DS* realization take 6-7 h, while for *snesim* the number is 2-3 h and for *iqsim* the number is 10-12 min. We emphasize that these times are for a specific setup, and that they will likely change for different configurations. Both the *snesim* and *DS* methods yield realizations with sufficient soft data conditioning, as
reflected by the low MHD variability of 0.48 and 0.78, respectively. The *iqsim* realizations showed a MHD variability of 1.79, which was due to insufficient soft data conditioning.

The presented practical example for modelling new survey areas uses a cognitive hydrostratigraphic model from one area as a TI to simulate the new area without a pre-existing cognitive model. The requirements are two-fold: 1) the geological settings of the two areas need to be similar and 2) the statistical hydrostratigraphic-petrophysical relationship needs to be
stationary between the two areas. The presented example shows a case where the two requirements are true, and the set of stochastic models are consistent with the cognitive geological model.

Finally, the importance of the underlying resistivity-hydrostratigraphic relationship has been shown. The relationship contains information on the translation of the continuous soft data variable into subsurface hydrostratigraphic units, and is indirectly used for soft data conditioning. The MPS modelling results are therefore sensitive towards the resistivity-
hydrostratigraphic relationship, and the more information acquired regarding the relationship, the better the realizations.

**Acknowledgements**. I would like to thank Jan Gunnink for introducing me to the world of geostatistics, and for sparking the interest that led to further research into Multiple-Point Statistics. This study is supported by HyGEM, Integrating geophysics, geology, and hydrology for improved groundwater and environmental management, project no. 11- 116763. The
funding for HyGEM is provided by The Danish Council for Strategic Research.



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
