# Peer review of "Hydrostratigraphic modeling using multiple-point statistics and airborne transient electromagnetic methods"

_Hydrology and Earth System Sciences, 2017_

## Referee Comment (RC1) · Anonymous Referee #1 · 26 Sep 2017

The paper applies and compares three multiple point statistics (MPS) methods (snesim, DS and iqsim) for hydrostratigraphic modelling using geological and geophysical data. This research is very relevant as (1) three MPS methods, including very recent methods, are compared to evaluate the advantages and disadvantages of each method which is very useful for users that want to select one of the different available MPS methods and (2) since these methods are all applied on a real-world case with realistic geological complexity and data availability.

The authors first apply the three MPS methods on their case study where the training image is actually identical to the model they want to simulate. This part is very extensive: the three MPS methods are used and different ways of validating the results are compared. The results of this part are according to me not so interesting since in a real case you never have the model you want to simulate but only one or more training images depicting some general geological concepts of the area. The results are also not surprising: iqsim better reproduces the TI which is logical since iqsim uses relatively large patches instead of pixels. In real cases, however, you don't want an exact reproduction of the TI but you want to simulate another area with similar patterns.

In the last part of their paper, the left half of the existing geological model is used as TI to simulate the right half of the model. For me, this second part is much more interesting. However, this part is very short: only one MPS method is used and different aspects of validation (such as comparison with boreholes) are not shown or discussed. I would like to see an application of the three MPS methods here and a more thorough description and discussion of the results as for the first part where the TI is equal to the result you want to obtain. For clarity and compactness of the paper, I would even propose to only do the full analysis on the second problem where another area is modelled and to remove the part where the TI is identical to the model.

Abstract, line 13 + introduction, lines 32-37: I would replace "hydrological" models by "hydrogeological models" or "groundwater models" as "hydrological" models could also refer to surface water modelling, rainfall-runoff modelling or river modelling which do not involve inclusion of geological and/or geophysical data.

---

## Referee Comment (RC2) · Anonymous Referee #2 · 18 Dec 2017

This paper provides an exhaustive comparison of three Multiple-Point-Statistics (MPS) methodologies - namely, Single normal equation simulation (snesim), Direct Sampling Simulation (DS) and Image quilting simulation (iqsim) - for the generation of random distributions of hydrofacies on a specific field site. For each methodology, the diverse realizations of hydrostratigraphic categories are obtained on the basis of 51 stochastically-reconstructed resistivity grids, to include the effect of uncertain conditioning (soft) data. The generated hydrostratigraphic models are compared against each other and against the Training Image (TI) (i) by visual inspection, (ii) in terms of the modified Hausdorff distance and (iii) in terms of the distance from borehole (hard) data. The paper is clearly written and the results will have wide application in the con-

text of field-scale stochastic facies reconstruction. I recommend the paper for publication in HESS, after that the authors address the questions/comments in the following itemized list:

- Advantages and disadvantages of each methodology are extensively discussed, and can be summarized as follows: (i) snesim is the best one in conditioning the simulations with soft data, thanks to the implicit Resistivity Atlas histograms. This methodology provides the best results in borehole distance for 2 out of 3 hydrostratigraphic categories. However, the resulting stochastics models are affected by unrealistic small scale variability, which implies a larger distance from the TI.

(ii) iqsim is the fastest algorithm amongst the three. It provides the smallest distance from the TI and the largest variability between realizations. On the other hand, it suffers from an improper conditioning from soft-data grids, as indicated by poor borehole-distance results.

(iii) DS is the most computationally expensive, it suffers from small-scale variability (line 556) and hydrostratigraphic units are not conditioned properly (line 753). It provides intermediate results in terms of all comparison metrics considered.

So, why did the authors choose DS as the unique methodology in the "Hydrostratigraphic modelling of new surveys", in Sect. 4.3? I would recommend to integrate this section also with the results of the other two methodologies for the simulation of "Area B".

- The absence of small-scale variability in single realization (iqsim) is regarded as an advantage. But, (1) as discussed in lines 733-741, this reconstructions can be regarded as the most realistic only if the TI is actually reproducing the correct scale of variability; (2) it is the model ensemble, and not the individual random realization, that is supposed to reflect the behavior of the whole system. Small-scale variations effect seem indeed to be reduced when evaluating the mode over the 10 realizations in sect. 4.3. The ensemble modes evaluated over each one of the three sets of 51 simulations

analyzed in the first part of the study should be also reported.

- It is not explored in this context how the three algorithms behave when generating random simulations with fixed conditioning data. What are the effects of the methods themselves on, e.g., the variability between realizations?

- line 458: " Here, sand & gravel and glacial clay were categorized into a single category, and hemipelagic clay was used as a background variable". The Modified Hausdorff distance is evaluated on binary images. Did the authors try to evaluate a MHD array separately for each category (similarly to what it is done for AEBD)?

- line 726: "The borehole distances of the iqsim realizations revealed exceedingly small hemipelagic clay distances, with average of 0.2 m"; line 730: "(...) the ample near surface hemipelagic clay decreses the hemipelagic clay borehole distance". If the presence of near-surface hemipelagic clay is an artifact of the algorithm (i.e. is not consistent with borehole data), why should it results in a decrease of the borehole distance?

- Figure 2: the figure caption and the references to the figure in the manuscript are not consistent with the letters (a-g) indicating the diverse frames of the picture.

- Eq. 2: Symbols $a\_i$ and $b\_i$ represent position vectors, but they are written as scalar quantities.

- line 536: "where $binlog\_i$ is the ith cell in the binary log grid" should be changed into "where $binlog\_i$ is the ith ACTIVE cell in the binary log grid".

---

## Author Comment (AC1) · 12 Jan 2018

**Response to referees, hess-2017-413**

Firstly, the authors would like to thank the two anonymous referees for taking their time to read the manuscript and providing detailed and constructive comments. The comments, questions and suggestions are addressed in the following sections.

**\*AC:** Author Comments
**\*RC:** Referee Comments

**Response to anonymous referee #1**

**General comments:**
'*The paper applies and compares three multiple point statistics (MPS) methods (snesim, DS and iqsim) for hydrostratigraphic modelling using geological and geophysical data. This research is very relevant as (1) three MPS methods, including very recent methods, are compared to evaluate the advantages and disadvantages of each method which is very useful for users that want to select one of the different available MPS methods and (2) since these methods are all applied on a real-world case with realistic geological complexity and data availability.*

**RC1:** '*The authors first apply the three MPS methods on their case study where the training image is actually identical to the model they want to simulate. This part is very extensive: the three MPS methods are used and different ways of validating the results are compared. The results of this part are according to me not so interesting since in a real case you never have the model you want to simulate but only one or more training images depicting some general geological concepts of the area. The results are also not surprising: iqsim better reproduces the TI which is logical since iqsim uses relatively large patches instead of pixels. In real cases, however, you don't want an exact reproduction of the TI but you want to simulate another area with similar patterns.*'

**AC1:** We disagree that from a practical point of view the first set of tests are not as interesting since the model we wish to simulate is usually not available. Instead, we consider this case a 'best case scenario' where the TI provides an accurate rendition of the 3D patterns relevant to the given model.
Some of the authors have recently submitted a research paper to HESS, which focuses on uncertainty related to the MPS setup, as well as a presentation of a more practical application, where a 3D geological model from another area is used as TI to simulate a hydrostratigraphic model using SkyTEM data and lithology logs (hess-2017-734; https://doi.org/10.5194/hess-2017-413).

**AC2:** Regarding the iqsim results we were actually surprised that they performed the best in resembling the cognitive geological model using the Modified Hausddorff distance (MHD) measure. If you look at the individual iqsim realizations (FIGURE 8D) you will quickly realize that the overall placement of the hydrostratigraphic units are not very precise, compared to snesim and DS (FIGURE 8B&C). In fact by looking at the vertical cross-sections the sporadic nature of the upper part of the realizations becomes clear, where the valleys (filled with 'sand & gravel' and 'glacial clay' can be covered by 'hemipelagic clay', which is not possible in the TI. Looking at the borehole distance results, iqsim has the highest average borehole distances for 'sand & gravel' and 'glacial clay' units.

**RC2:** '*In the last part of their paper, the left half of the existing geological model is used as a TI to simulate the right half of the model. For me, this second part is much more interesting. However, this part is very short: only*

*one MPS method is used and different aspects of validation (such as comparison with boreholes) are not shown or discussed. I would like to see an application of the three MPS methods here and a more thorough description and discussion of the results as for the first part where the TI is equal to the result you want to obtain. For clarity and compactness of the paper, I would even propose to only do the full analysis on the second problem where another area is modeled and to remove the part where the TI is identical to the model.'*

**AC3:** As stated above in **AC1**, *w*e disagree with the statement that the second part is "more" interesting than the first part. Again, we consider the first case a *"pretend case" where* the TI contains the actual 3D patterns of the target model and is therefore a "best case scenario".
Furthermore, focusing on the second case, the half-sim case, a problem occurs since the TI is suddenly cut in half. Cutting the TI in half results in a reduction of the patterns contained in the TI and, as discussed by e.g. Emery and Lantuéjoul (2014), if the size of the patterns contained in the TI are too small we do not properly reproduce the desired patterns. If they become too small the information is simply not available. In this case the valleys are cut in half and only part of the valley structures are present in the half TI. Therefore, although the second half sim case is more interesting from a practical point of view, it is simply not an ideal setup for making such tests.
A recent paper has been submitted to HESS (hess-2017-734; https://doi.org/10.5194/hess-2017-413) where snesim is used to test if we can use 3D hydrostratigraphic models from a different survey area to create 3D hydrostratigraphic models.
The exact same half sim case, with the exact same data, was also presented for iqsim by Hoffimann *et al.* (2017). We will add a reference to Hoffimann *et al.* (2017) in the section describing the half-sim case and include the results by Hoffimann *et al.* (2017) in the discussion of the half-sim case.

**RC3:** '*Abstract, line 13 + introduction, lines 32-37: I would replace "hydrological" models by "hydrogeological models" or "groundwater models" as "hydrological" models could also refer to surface water modelling, rainfall-runoff modelling or river modelling which do not involve inclusion of geological and/or geophysical data.'*

**AC4:** Good point, we will change that during the revision.

**Response to anonymous referee #2**

**General comments:**
'*This paper provides an exhaustive comparison of three Multiple-Point-Statistics (MPS) methodologies - namely, Single normal equation simulation (snesim), Direct Sampling Simulation (DS) and Image quilting simulation (iqsim) - for the generation of random distributions of hydrofacies on a specific field site. For each methodology, the diverse realizations of hydrostratigraphic categories are obtained on the basis of 51 stochastically-reconstructed resistivity grids, to include the effect of uncertain conditioning (soft) data. The generated hydrostratigraphic models are compared against each other and against the Training Image (TI) (i) by visual inspection, (ii) in terms of the modified Hausdorff distance and (iii) in terms of the distance from borehole (hard) data. The paper is clearly written and the results will have wide application in the con-text of field-scale stochastic facies reconstruction. I recommend the paper for publication in HESS, after that the authors address the questions/comments in the following itemized list.*'

**RC1:** '*Advantages and disadvantages of each methodology are extensively discussed, and can be summarized as follows:*
> *(i) snesim is the best one in conditioning the simulations with soft data, thanks to the implicit Resistivity Atlas histograms. This methodology provides the best results in borehole distance for 2 out of 3 hydrostratigraphic categories. However, the resulting stochastics models are affected by unrealistic small scale variability, which implies a larger distance from the TI.*
> *(ii) iqsim is the fastest algorithm amongst the three. It provides the smallest distance from the TI and the largest variability between realizations. On the other hand, it suffers from an improper conditioning from soft-data grids, as indicated by poor borehole distance results.*
> *(iii) DS is the most computationally expensive, it suffers from small-scale variability (line 56) and hydrostratigraphic units are not conditioned properly (line 753). It provides intermediate results in terms of all comparison metrics considered.*

*So, why did the authors choose DS as the unique methodology in the "Hydrostratigraphic modelling of new surveys", in Sect. 4.3? I would recommend to integrate this section also with the results of the other two methodologies for the simulation of "Area B".*'

**AC1:** The choice of method was not crucial here since the "Hydrostratigraphic modelling of new surveys" was only meant as an example of a practical application of MPS in relation to 3D hydrostratigraphic voxel modelling. Since the DS method is easy to parameterize and easy to setup for running in parallel on a computer cluster it was chosen over using the SGeMS implementation of snesim. This should probably be mentioned in the revised paper.
Regarding integrating the other methods in Sect. 4.3 see author comment 3 and 4 in the response to anonymous referee #1.

**RC2:** '*The absence of small-scale variability in single realization (iqsim) is regarded as an advantage. But, (1) as discussed in lines 733-741, this reconstructions can be regarded as the most realistic only if the TI is actually reproducing the correct scale of variability; (2) it is the model ensemble, and not the individual random realization, that is supposed to reflect the behavior of the whole system. Small-scale variations effect seem indeed to be reduced when evaluating the mode over the 10 realizations in sect. 4.3. The ensemble modes evaluated over each one of the three sets of 51 simulations analyzed in the first part of the study should be also reported.*'

**AC2:** The absence of small-scale variability of a single realization seems to be part of the reason for the smaller $MHD_{cog}$-distances in the iqsim realizations. So in comparison with the TI, which does not contain small-scale

variability, the iqsim realizations are the most similar, not realistic. We will revise the text so that it is clear that small-scale variability does not mean less realistic realizations. Furthermore, a new figure presenting the ensemble modes for each of the three algorithms will be considered strongly for the final draft.

**RC3:** '*It is not explored in this context how the three algorithms behave when generating random simulations with fixed conditioning data. What are the effects of the methods themselves on, e.g., the variability between realizations?*'

**AC3:** We are not sure what the referee means by "generating random realizations with fixed conditioning data". We assume what is meant is to run realizations with borehole data as hard conditioning data. The usage of hard borehole data for conditioning was not important for the goal of this paper, which was focused on comparing MPS algorithms using an extensive soft SkyTEM data set. However, a recent paper has been submitted to HESS (hess-2017-734), where snesim realizations are conditioned to both soft SkyTEM data and hard borehole data.

**RC4:** '*line 458: "Here, sand & gravel and glacial clay were categorized into a single category, and hemipelagic clay was used as a background variable". The Modified Hausdorff distance is evaluated on binary images. Did the authors try to evaluate a MHD array separately for each category (similarly to what it is done for AEBD)?*'

**AC4:** This is a good observation, and should probably be stated more clearly in the revised manuscript why we make an evaluation based on binary images. The reason for this was the computational overhead of computing the Modified Hausdorff Distance for 51 models containing 1,187,823 cells (229x133x39). Even after representing the geometric objects of each realization as outlines only, the computational burden was still too large for computing the MHD for each separate category.

**RC5:** '*line 726: "The borehole distances of the iqsim realizations revealed exceedingly small hemipelagic clay distances, with average of 0.2 m"; line 730: "(...) the ample near surface hemipelagic clay decreases the hemipelagic clay borehole distance". If the presence of near-surface hemipelagic clay is an artifact of the algorithm (i.e. is not consistent with borehole data), why should it results in a decrease of the borehole distance?*'

**AC5:** The text will be revised and should instead reflect that if hemipelagic clay is present at the surface then the average MHD increases, and the reason for the low hemipelagic clay distances should be found elsewhere. Instead, in the revised paper, it will be made clear that there is a trade-off relationship between the borehole distances of each of the three lithological categories. In the iqsim case the average distance is low for hemipelagic clay (0.2 m) while increased for the glacial clay (3.5 m) and sand & gravel (5.8 m) categories. The summed distance of the three lithological categories is therefore 9.5 m for iqsim, while the summed distance for DS is 8.5 m and for snesim the distance is 7.5 m.

**RC6:** '*Figure 2: the figure caption and the references to the figure in the manuscript are not consistent with the letters (a-g) indicating the diverse frames of the picture.*'

*AC6: This is not intended and the figure caption will be edited so it corresponds to the actual figure.*

**RC7:** '*Eq. 2: Symbols a_i and b_i represent position vectors, but they are written as scalar quantities.*'

*AC7: The symbols will be italic to indicate that they are vectors in the revised version.*

**RC8:** '*line 536: "where binlog_i is the ith cell in the binary log grid" should be changed into "where binlog_i is the ith ACTIVE cell in the binary log grid".*'

*AC8:* *Noted, the text will be edited for the final draft.*

---

## Author Response (AR1)

Dear Mrs. Riva,

We have decided to revise the paper according to the comments from the two anonymous reviewers. It should be mentioned that the author list has been rearranged so the degree of contribution of the different co-authors is reflected accordingly, which was not previously the case.

In particular, the paper has been revised in regards to both referees mentioning their interest in expanding the section related to the section 4.3. Emphasis on the significance of the test case has been increased, while the issue related to reducing the size of the TI are described in the discussion. A list of miscellaneous corrections has also been composed and can be found in the "point-by-point reply" found below. Regarding the considerations of implementing a new figure with the ensemble averages of each of the 51 realizations for the different MPS methods, we chose to not include this figure since it did not change the conclusions of the paper.

Kind regards, Adrian A.S. Barfod

**Point-by-point reply, hess-2017-413**

Firstly, the authors would like to thank the two anonymous referees for taking their time to read the manuscript and providing detailed and constructive comments. The comments, questions and suggestions are addressed in the following sections.

**\*AC:** Author Comments **\*RC:** Referee Comments**

NOTE: All of the line numbers in this document refer to the revised manuscript and not the "manuscript changes" document

**Response to anonymous referee #1**

**General comments:**

`The paper applies and compares three multiple point statistics (MPS) methods (snesim, DS and iqsim) for hydrostratigraphic modelling using geological and geophysical data. This research is very relevant as (1) three MPS methods, including very recent methods, are compared to evaluate the advantages and disadvantages of each method which is very useful for users that want to select one of the different available MPS methods and (2) since these methods are all applied on a real-world case with realistic geological complexity and data availability.'

**RC1:** 'The authors first apply the three MPS methods on their case study where the training image is actually identical to the model they want to simulate. This part is very extensive: the three MPS methods are used and different ways of validating the results are compared. The results of this part are according to me not so interesting since in a real case you never have the model you want to simulate but only one or more training images depicting some general geological concepts of the area. The results are also not surprising: iqsim better reproduces the TI which is logical since iqsim uses relatively large patches instead of pixels. In real cases, however, you don't want an exact reproduction of the TI but you want to simulate another area with similar patterns.'

**AC1:** We disagree that from a practical point of view the first set of tests are not as interesting since the model we wish to simulate is usually not available. Instead, we consider this case a "pseudo synthetic test case" where the TI provides an accurate rendition of the 3D patterns relevant to the given model.

Regarding the concern that you would never have the model you wish to simulate available to use as a TI, this is simply a case where the TI, synthetic or an actual real-world model, is a precise rendition of the expected hydrostratigraphic architecture.

Some of the authors have recently submitted a research paper to HESS, which is currently in open discussion. The paper focuses on the uncertainty related to the MPS setup, as well as a presentation of a more practical application, where a 3D geological model from another area is used as TI to simulate a hydrostratigraphic model using SkyTEM data and lithology logs (hess-2017-734; https://doi.org/10.5194/hess-2017-734).

**AC2:** Regarding the iqsim results, we were actually quite surprised that they performed the best in resembling the cognitive geological model based on the Modified Hausddorff distance ( $D_{MH}$ ). If the individual iqsim realizations are studied (FIGURE 8D) it is seen that the overall placement of the hydrostratigraphic units are not very precise, compared to snesim and DS (FIGURE 8B&C). In fact by looking at the vertical cross-sections the sporadic nature of the upper part of the realizations becomes clear, where the valleys (filled with 'sand & gravel'

and 'glacial clay' can be covered by 'hemipelagic clay', which is not possible in the TI. Looking at the borehole distance results, iqsim has the highest average borehole distances for 'sand & gravel' and 'glacial clay' units.

**Resulting corrections:**

Line 18-19: The main case is introduced as a "synthetic test case", so that it is made clear that it is a controlled environment in which we can test the different MPS methods against each other

Line 116: the halfsim case is no longer referred to as a "practical example"

Line 120-122: The main case is introduced as a "synthetic test case"

Line 128: the halfsim case is no longer referred to as a practical example, but instead just as an example Table 4 – "row average" column added which relates to the borehole distances, and describes the trade-off relationship between the borehole distances respective of the individual hydrostratigraphic units. Line 639: The "row average" column of Table 4 is referred to in the text

**RC2:** 'In the last part of their paper, the left half of the existing geological model is used as a TI to simulate the right half of the model. For me, this second part is much more interesting. However, this part is very short: only one MPS method is used and different aspects of validation (such as comparison with boreholes) are not shown or discussed. I would like to see an application of the three MPS methods here and a more thorough description and discussion of the results as for the first part where the TI is equal to the result you want to obtain. For clarity and compactness of the paper, I would even propose to only do the full analysis on the second problem where another area is modeled and to remove the part where the TI is identical to the model.'

**AC3:** As stated above in **AC1**, we disagree with the statement that the second part is "more" interesting than the first part. Again, we consider the first case a "pretend case" where the TI contains the actual 3D patterns of the target model and is therefore a "best case scenario".**

Furthermore, focusing on the second case, the half-sim case, a problem occurs since the TI is cut in half. Cutting the TI in half, results in a reduction of the patterns contained in the TI, and, as discussed by e.g. Emery and Lantuéjoul (2014), if the size of the patterns contained in the TI are too small we do not properly reproduce the desired patterns. If they become too small the information is simply not available. In this case some of the valleys are cut in half and only part of the valley structures are present in the half TI. Therefore, although the second half sim case is interesting from a practical point of view, it is simply not an ideal setup for making such tests. The discussion text has been edited to reflect these issues.

A recent paper has been submitted to HESS (hess-2017-734; https://doi.org/10.5194/hess-2017-734) and is currently in the public discussion phase. In the paper, snesim is used to test if it is possible to use a 3D hydrostratigraphic models from a different survey area to create 3D hydrostratigraphic models. The exact same half sim case, with the exact same data, was also presented for iqsim by Hoffimann *et al.* (2017) We will add a reference to Hoffimann *et al.*

(2017). We will add a reference to Hoffimann *et al.* (2017) in the section describing the half-sim case and include the results by Hoffimann *et al.* (2017) in the discussion of the half-sim case.

**Resulting corrections:**

Line 777-782: The issue related to cutting patterns from the TI is addressed. Line 680-681: The Hoffimann *et al.* (2017) study is mentioned

**RC3:** 'Abstract, line 13 + introduction, lines 32-37: I would replace "hydrological" models by "hydrogeological models" or "groundwater models" as "hydrological" models could also refer to surface water modelling, rainfall-runoff modelling or river modelling which do not involve inclusion of geological and/or geophysical data.'

AC4: Good point, this has been edited

**Resulting corrections:**

Line 13: "hydrological models" has been replaced with "groundwater models" Line 34: "hydrological modeling" has been replaced with groundwater modeling Line 35: "hydrological modeling" has been replaced with groundwater modeling

**Response to anonymous referee #2**

**General comments:**

`This paper provides an exhaustive comparison of three Multiple-Point-Statistics (MPS) methodologies - namely, Single normal equation simulation (snesim), Direct Sampling Simulation (DS) and Image quilting simulation (iqsim) - for the generation of random distributions of hydrofacies on a specific field site. For each methodology, the diverse realizations of hydrostratigraphic categories are obtained on the basis of 51 stochasticallyreconstructed resistivity grids, to include the effect of uncertain conditioning (soft) data. The generated hydrostratigraphic models are compared against each other and against the Training Image (TI) (i) by visual inspection, (ii) in terms of the modified Hausdorff distance and (iii) in terms of the distance from borehole (hard) data. The paper is clearly written and the results will have wide application in the con-text of field-scale stochastic facies reconstruction. I recommend the paper for publication in HESS, after that the authors address the questions/comments in the following itemized list.'

**RC1:** *Advantages and disadvantages of each methodology are extensively discussed, and can be summarized as follows:*

(i) snesim is the best one in conditioning the simulations with soft data, thanks to the implicit Resistivity Atlas histograms. This methodology provides the best results in borehole distance for 2 out of 3 hydrostratigraphic categories. However, the resulting stochastics models are affected by unrealistic small scale variability, which implies a larger distance from the TI.

(ii) iqsim is the fastest algorithm amongst the three. It provides the smallest distance from the TI and the largest variability between realizations. On the other hand, it suffers from an improper conditioning from soft-data grids, as indicated by poor borehole distance results.

(iii) DS is the most computationally expensive, it suffers from small-scale variability (line 56) and hydrostratigraphic units are not conditioned properly (line 753). It provides intermediate results in terms of all comparison metrics considered.

*So, why did the authors choose DS as the unique methodology in the "Hydrostratigraphic modelling of new surveys", in Sect. 4.3? I would recommend to integrate this section also with the results of the other two methodologies for the simulation of "Area B"."*

**AC1:** The choice of method was not crucial here since the "Hydrostratigraphic modelling of new surveys" was only meant as an experiment to portray the practical applications of MPS in relation to 3D hydrostratigraphic voxel modelling. Since the DS method is easy to parameterize and easy to setup for running in parallel on a computer cluster it was chosen over using the SGeMS implementation of snesim. This is now reflected in the paper.

Regarding integrating the other methods in Sect. 4.3, see author comment 3 and 4 in the response to anonymous referee #1.

**Resulting corrections:**

Line 678-681: the choice of the DS method for the halfsim case is elaborated

**RC2:** 'The absence of small-scale variability in single realization (iqsim) is regarded as an advantage. But, (1) as discussed in lines 733-741, this reconstructions can be regarded as the most realistic only if the TI is actually reproducing the correct scale of variability; (2) it is the model ensemble, and not the individual random

realization, that is supposed to reflect the behavior of the whole system. Small-scale variations effect seem indeed to be reduced when evaluating the mode over the 10 realizations in sect. 4.3. The ensemble modes evaluated over each one of the three sets of 51 simulations analyzed in the first part of the study should be also reported.'

**AC2:** The absence of small-scale variability of a single realization seems to be part of the reason for the smaller MHDcog-distances in the iqsim realizations. So, in comparison with the TI, which does not contain small-scale variability, the iqsim realizations are the most similar, not realistic. The text has been revised to make it clear that small-scale variability does not mean less realistic realizations. Furthermore, a new figure presenting the ensemble modes for each of the three algorithms will be considered strongly for the final draft.

**Resulting corrections:**

Line 550, 551, 562, 563, 688 and 756: The usage of the word "unrealistic" in tandem with "short scale variability" is avoided

The ensemble modes for each of the three algorithms was not implemented since it did not change the results and conclusions of the paper.

**RC3:** 'It is not explored in this context how the three algorithms behave when generating random simulations with fixed conditioning data. What are the effects of the methods themselves on, e.g., the variability between realizations?'

**AC3:** We are not sure what the referee means by "generating random realizations with fixed conditioning data". We assume what is meant is to run realizations with borehole data as hard conditioning data. The usage of hard borehole data for conditioning was not important for the goal of this paper, which was focused on comparing MPS algorithms using an extensive soft SkyTEM data set. However, a recent paper has been submitted to HESS (hess-2017-734), where snesim realizations are conditioned to both soft SkyTEM data and hard borehole data.

**RC4:** *`line 458: "Here, sand & gravel and glacial clay were categorized into a single category, and hemipelagic clay was used as a background variable". The Modified Hausdorff distance is evaluated on binary images. Did the authors try to evaluate a MHD array separately for each category (similarly to what it is done for AEBD)?'*

**AC4:** This is a good observation, and in the revised paper it is stated more clearly why we make an evaluation based on binary images. The reason for this was the computational overhead of computing the Modified Hausdorff Distance for 51 models containing 1,187,823 cells (229x133x39). Even after representing the geometric objects of each realization as outlines only, the computational burden was still too large for computing the DMH for each separate category.

**Resulting corrections:**

Line 454-457: computational burden related to DMH is mentioned Line 594-595: The combining of *glacial clay* and *sand & gravel* units is mentioned Line 599: The binary classification of the realizations is emphasized Line 602: The fact that the DMH simply measures the location of the valleys due to the binary classification of the realizations is emphasized

**RC5:** *'line 726: "The borehole distances of the iqsim realizations revealed exceedingly small hemipelagic clay distances, with average of 0.2 m"; line 730: "(...) the ample near surface hemipelagic clay decreases the hemipelagic clay borehole distance". If the presence of near-surface hemipelagic clay is an artifact of the algorithm (i.e. is not consistent with borehole data), why should it results in a decrease of the borehole distance?'*

**AC5:** The text has been revised and now reflects that if hemipelagic clay is present at the surface then the average  $D_{MH}$  increases, and the reason for the low hemipelagic clay distances should be found elsewhere. Instead, in the revised paper, it is made clear that there is a trade-off relationship between the borehole distances of each of the three lithological categories. In the iqsim case the average distance is low for hemipelagic clay (0.2 m) while increased for the glacial clay (3.5 m) and sand & gravel (5.8 m) categories. The row average of the three lithological categories is therefore 9.5 m for iqsim, while the summed distance for DS is 8.5 m and for snesim the distance is 7.5 m.

**Resulting corrections:**

Table 4 – "row average" column added Line 748-756: The trade-off relationship is elaborated upon

**RC6:** *Figure 2: the figure caption and the references to the figure in the manuscript are not consistent with the letters (a-g) indicating the diverse frames of the picture.*

**AC6:** This is not intended and the figure caption has been edited so it corresponds to the actual figure.

**Resulting corrections:** Figure 2: caption has been edited**

**RC7:** *`Eq. 2: Symbols a\_i and b\_i represent position vectors, but they are written as scalar quantities.'*

**AC7:** The symbols are now italic to indicate that they are vectors in the revised version.

**Resulting corrections:**

Equation 2: corrected

**RC8:** `*line 536: "where binlog\_i is the ith cell in the binary log grid" should be changed into "where binlog\_i is the ith ACTIVE cell in the binary log grid".*'

**AC8: Noted, the text has been edited**

**Resulting corrections:**

Line 531: The word "active" has been inserted

**Miscellaneous corrections:**

- The author list has been changed, since the order was previously not representative of the degree of contribution of the different co-authors
- The Høyer *et al.* 2016 reference has been updated since the paper has now official been published
- An additional reference to the cognitive modeling approach has been added (Line 44)
- "Modelling" has been corrected to "modeling" throughout the revised paper to correspond with grammar rules related to writing English (United Stated)
- MHD has been changed to DMH throughout the revised paper in accordance with the HESS guidelines
- AEBD has been changed to DAEB throughout the revised paper in accordance with the HESS guidelines
- The following words are no longer written in *Italics*: snesim, DS and iqsim.

- In Figure 7 the reference to meltwater silt has been corrected to meltwater sand
- The final paragraph of the discussion was removed since it did not really discuss anything in particular
- The discussion of short scale variability and the usage of post-processing has been removed from the last paragraph of section 4.3, since post-processing is generally not considered in this paper.

**Hydrostratigraphic modellingmodeling using multiple-point statistics and airborne transient electromagnetic methods**

Adrian A.S. Barfod1,2, Julien Straubhaar4, Ingelise Møller1, Anders V. Christiansen2, Anne-Sophie Høyer1, Júlio Hoffimann3, Anders V. Christiansen2, Ingelise Møller4Julien Straubhaar4, Jef Caers3

 1Department of Groundwater and Quaternary Geology Mapping, Geological Survey of Denmark and Greenland (GEUS), C.F. Møllers Allé 8, 8000 Aarhus C
 2Hydrogeophysics group, Department of Geoscience, Aarhus University, C.F. Møllers Allé 4, 8000 Aarhus C
 3Stanford Center for Reservoir Forecasting, School of Earth, Energy & Environmental Sciences, Stanford University, Green Earth Sciences, 367 Panama St, Stanford, CA 94305

10 4Centre d'Hydrogéologie et de Géothermie (CHYN), Université de Neuchâtel, Switzerland

Correspondence to: Adrian A.S. Barfod (adrian.s.barfod@gmail.com)

Abstract. Creating increasingly realistic hydrologicalgroundwater models involves the inclusion of additional geological and geophysical data in the hydrostratigraphic modellingmodeling procedure. Using Multiple Point Statistics (MPS) for stochastic
 hydrostratigraphic modellingmodeling provides a degree of flexibility that allows the incorporation of elaborate datasets and

- provides a framework for stochastic hydrostratigraphic modellingmodeling. This paper focuses on comparing three MPS methods: snesim, DS and iqsim. The MPS methods are tested and compared on a real-world hydrogeophysical survey from Kasted in Denmark, which covers an area of 45 km2. A controlled test environment, similar to a synthetic test case, is constructed from the Kasted survey and is used to compare the modeling results of the three aforementioned MPS methods.
- 20 The comparison of the stochastic hydrostratigraphic MPS models is carried out in an elaborate scheme of visual inspection, mathematical similarity and consistency with boreholes. Using the Kasted survey data, a practicalan example for modellingmodeling new survey areas is presented. A cognitive hydrostratigraphic model of one area is used as Training Image to create a suite of stochastic hydrostratigraphic models in a new survey area. The advantage of stochastic modellingmodeling is that detailed multiple point information from one area can be easily transferred to another area considering uncertainty.
- 25 The presented MPS methods each have their own set of advantages and disadvantages. The DS method had average computation times of 6-7 h, which is large, compared to jqsim with average computation times of 10-12 min. However, jqsim generally did not properly constrain the near-surface part of the spatially dense soft data variable. The computation time of 2-3 h for snesim was in between DS and jqsim. The snesim implementation used here is part of the Stanford Geostatistical Modeling Software, or SGeMS. The snesim setup was not trivial, with numerous parameter settings, usage of multiple grids and a search tree database. However, once the parameters had been set it yielded comparable results to the other methods. Both, jqsim and DS, are easy to script and run in parallel on a server, which is not the case for the snesim implementation in SGeMS.

**1** Introduction**

Recent advances in hydrologygroundwater modeling have shown the importance of accurate hydrologiehydrogeologic models
for management of increasingly sparse groundwater resources. Hydrologic modellingGroundwater modeling predictions are sensitive to geologic heterogeneity (e.g. Freeze 1975, Gelhar 1984, Fogg et al. 1998, LaBolle and Fogg 2001, Zheng and Gorelick 2003, Feyen and Caers 2006, Fleckenstein et al. 2006, Zhao and Illman 2017). However, geological units include complexities not directly related to hydrofacies (Klingbeil et al. 1999). Instead, the concept of *hydrostratigraphic units* is used throughout this study, which effectively combines geological units and reduces the total number of units resulting in a closer
relation to the hydrologic units. Improving the realism and quantification of uncertainty around hydrostratigraphic models is

[revised manuscript text omitted]

(2011). This bypasses the necessity of saving spatial patterns in a search tree database; instead2 spatial patterns are conditioned 230 by directly scanning the TI.

One issue which needs to be solved is how to constrain a soft data variable. In DS, this is accomplished by introducing an auxiliary variable. The auxiliary variable is roughly a translation of the TI into a soft data variable. Suppose a forward operator, denoted by *G*, represents the physical model, which translates the subsurface hydrostratigraphic units into the continuous soft data variable, as when scanning the near surface with a geophysical instrument and subsequently process and interpret the data 235 into the actual petrophysical parameter. Then we can define an approximate forward operator G\* (Mariethoz and Caers 2014b).

- The  $G^*$  operator is an operator which is used to translate the TI into a spatially overlapping soft data variable. However, in practice creating a  $G^*$  operator requires several steps. Based on the modeling setup of this study, we will briefly review the required steps. Firstly, the TI needs to be populated with relevant resistivity values. The resulting populated resistivity grid does, however, not reflect the physical model, G, which translates the subsurface hydrostratigraphic units into subsurface bulk
- 240 resistivity. To properly reflect the G operator additional complexity needs to added, such as: smooth layer boundaries, loss of resolution with depth, limited resolution capabilities, the instrument footprint *etc*. This can be achieved by using either an approximate 1D or a full 3D forward modeling code to translate the populated resistivity models into synthetic data reflecting actually measured field data. These data, the forward responses then need to be processed and inverted back to resistivity models, which now constitute an auxiliary variable, which reflects the complexities involved with the SkyTEM system. The
- 245 auxiliary variable and the categorical hydrostratigraphic variable are combined to create a multivariate, or bivariate TI. The bivariate TI consists of a categorical variable, *e.g.* the three hydrostratigraphic units, and the geographically overlapping continuous auxiliary variable, representing the soft data variable. The setup used in this paper, avoids the usage of the G\* operator to create the auxiliary variable, since the reconstructed resistivity grids and cognitive hydrostratigraphic model grids geographically overlap. The reconstructed resistivity grids can thus directly be used as an auxiliary variable for the cognitive
- 250 hydrostratigraphic model TI. The bivariate TI constituted of collocated categorical hydrostratigraphic units (cognitive model / primary variable) and resistivity values (auxiliary variable) contains information regarding the relationship between these variables. The simulation is then conditioned against the bivariate TI by using a so-called distance measure. Distance measures are designed to compare the similarity of two sets of spatial patterns to each other. The idea is that similar patterns have relatively small distances, while dissimilar patterns have relatively large distance values. Conditioning against the MP information contained in the bivariate TI enables the ability to find probable spatial patterns, which also agree with the soft data variable.

DS is more flexible than traditional MPS methods, such as snesim. As no search-tree database is required, the multiple grid formulation used in *snesim* is not required in DS, which effectively reduces the number of parameters and makes the parametrization relatively simple. Furthermore, one can simulate continuous variables, and/or discrete variables with no limitation to the maximum number of categories (*e.g.* hydrostratigraphic units). In our case, any number of geophysical datasets collocated or not, can be included as long as a corresponding auxiliary variable is added to the multivariate TI. However, it

can be a cumbersome process generating the auxiliary variable. Furthermore, it is even possible to use probability grids in place of the actual soft data variable, as in snesim, if desired (Mariethoz et al. 2015). Depending on the setup and dataset, DS can be computationally as fast as snesim. Moreover, the DS implementation used in this work is amenable to scripting yielding
 the possibility of improving computation times on computer clusters or servers, if available.

**3.1.3 Image quilting simulation - iqsim**

The image quilting simulation (iqsim) method has been borrowed from the computer vision literature (Efros and Freeman 2001). The algorithm is originally designed to synthesize and/or replicate patterns from 2D images, but has since been modified to accommodate conditioning data and 3D geoscience problems (Mahmud et al. 2014). The concept of the iqsim method is

[revised manuscript text omitted]

<sup>1 Software is available at https://github.com/juliohm/ImageQuilting.jl.

The uncertainty related to the stochastic resistivity grids is different from the Kriging resistivity grid uncertainty. The standard deviation (STD) related to the Kriging reconstructed grid is closely related to the distance to the nearest data point (Figure 2d),
whereas the uncertainty on the stochastic resistivity grids reveals values much more correlated to the patterns of the geophysical information.